# A bacterial membrane sculpting protein with BAR domain-like activity

Daniel A Phillips[1,2]*[†], Lori A Zacharoff[3]*[†], Cheri M Hampton[4], Grace W Chong[5], Anthony P Malanoski[6], Lauren Ann Metskas[1], Shuai Xu[3], Lina J Bird[6], Brian J Eddie[6], Aleksandr E Miklos[1], Grant J Jensen[7,8], Lawrence F Drummy[4], Mohamed Y El-Naggar[9], Sarah M Glaven[6]

[1]U.S. Army DEVCOM Chemical Biological Center, BioSciences Division, BioChemistry Branch, Aberdeen Proving Ground, United States; [2]Oak Ridge Institute for Science and Education, Oak Ridge, United States; [3]University of Southern California, Department of Physics and Astronomy, Los Angeles, United States; [4]Materials and Manufacturing Directorate, Air Force Research Laboratory, Wright-Patterson Air Force Base, Dayton, United States; [5]University of Southern California, Department of Biological Sciences, Los Angeles, United States; [6]Center for Bio/Molecular Science and Engineering, Naval Research Laboratory, Washington, United States; [7]California Institute of Technology, Division of Biology and Biological Engineering, Pasadena, United States; [8]Department of Chemistry and Biochemistry, Brigham Young University, Provo, United States; [9]University of Southern California, Department of Chemistry, Los Angeles, United States

*For correspondence:
daniel.a.phillips62.ctr@army.mil
(DAP);
zacharoff_lori@bah.com (LAZ)

[†]These authors contributed
equally to this work

Competing interest: See page
22

Reviewing Editor: Karina B
Xavier, Instituto Gulbenkian de
Ciência, Portugal

**Abstract** Bin/Amphiphysin/RVS (BAR) domain proteins belong to a superfamily of coiled-coil proteins influencing membrane curvature in eukaryotes and are associated with vesicle biogenesis, vesicle-mediated protein trafficking, and intracellular signaling. Here, we report a bacterial protein with BAR domain-like activity, BdpA, from *Shewanella oneidensis* MR-1, known to produce redox-active membrane vesicles and micrometer-scale outer membrane extensions (OMEs). BdpA is required for uniform size distribution of membrane vesicles and influences scaffolding of OMEs into a consistent diameter and curvature. Cryo-TEM reveals that a strain lacking BdpA produces lobed, disordered OMEs rather than membrane tubules or narrow chains produced by the wild-type strain. Overexpression of BdpA promotes OME formation during planktonic growth of *S. oneidensis* where they are not typically observed. Heterologous expression results in OME production in *Marinobacter atlanticus* and *Escherichia coli*. Based on the ability of BdpA to alter membrane architecture in vivo, we propose that BdpA and its homologs comprise a newly identified class of bacterial BAR domain-like proteins.

## Introduction

Bacterial outer membrane vesicle (OMV) formation is ubiquitous and has many documented functions (*Toyofuku et al., 2018*). Outer membrane extensions (OMEs) are less commonly observed. OMEs remain attached to the cell, and various morphologies can be seen extending from single cells including *Myxococcus xanthus* (*Remis et al., 2014*; *Wei et al., 2014*), flavobacterium strain Hel3_A1_48 (*Fischer et al., 2019*), *Vibrio vulnificus* (*Hampton et al., 2017*), *Francisella novicida* (*Sampath et al., 2018*), as cell-cell connections in *Bacillus subtilis* (*Bhattacharya et al., 2019*; *Dubey et al., 2016*; *Dubey and Ben-Yehuda, 2011*), and *Escherichia coli* (*Pande et al., 2015*), and as redox-active nanowires in *Shewanella oneidensis* (*Chong et al., 2019*; *Subramanian et al., 2018*; *Pirbadian et al., 2014*; *Gorby et al., 2006*). Researchers suspect that pathways for OMV and OME formation

have some mechanistic overlap (*Fischer et al., 2019*). Bacterial cell membrane curvature is observed during the formation of OMVs and OMEs, and it is proposed that proteins are necessary to stabilize these structures (*Bohuszewicz et al., 2016*). Several bacterial proteins have demonstrated membrane tubule formation capabilities in vitro (*Tanaka et al., 2010*; *Danne et al., 2017a*; *Danne et al., 2017b*; *Danne et al., 2015*; *Low et al., 2009*; *Low and Löwe, 2006*), but despite the growing number of reports, proteins involved in shaping bacterial membranes into OMV/OMEs in living cells have yet to be identified.

In eukaryotes, Bin/Amphiphysin/Rvs (BAR) domain-containing proteins generate membrane curvature through electrostatic interactions between positively charged amino acids and negatively charged lipids scaffolding the membrane along the intrinsically curved surface of the antiparallel coiled-coil protein dimers (*Peter et al., 2004*; *Frost et al., 2007*; *Shimada et al., 2007*; *Weissenhorn, 2005*). The extent of accumulation of BAR domain proteins at a specific site can influence the degree of the resultant membrane curvature (*Simunovic et al., 2015*), and tubulation events arise as a consequence of BAR domain multimerization in conjunction with lipid binding (*Mim et al., 2012*). Interactions between BAR domain proteins and membranes resolve membrane tension, promote membrane stability, and aid in localizing cellular processes, such as actin binding, signaling through small GTPases, membrane vesicle scission, and vesicular transport of proteins (*Habermann, 2004*; *Miki et al., 2000*; *Carman and Dominguez, 2018*). Some BAR domain-containing proteins, such as the N-BAR protein BIN1, contain an amphipathic alpha helical wedge that inserts into the outer membrane leaflet and can assist in BAR domain binding to the target membrane (*Drin and Antonny, 2010*). Other BAR domains can be accompanied by a membrane targeting domain, such as PX for phosphoinositide binding (*Seet and Hong, 2006*; *Itoh and De Camilli, 2006*), in order to direct membrane curvature formation at specific sites, as is the case with sorting nexin BAR (SNX-BAR) proteins (*Knævelsrud et al., 2013*). These SNX-BAR domain proteins involved in endocytic and vesicle transport mechanisms can be effector targets during bacterial infections (*Elwell et al., 2017*; *Paul et al., 2017*; *Mirrashidi et al., 2015*; *Aeberhard et al., 2015*; *Latomanski et al., 2016*; *Liebl et al., 2017*). Likewise, F-BAR protein PACSIN2, known to remodel cell membranes and associate with the actin cytoskeleton, is specifically recruited during HIV-1 infection and aids in cell-to-cell spreading of viral particles (*Popov et al., 2018*).

Despite our knowledge of numerous eukaryotic BAR domain-containing proteins spanning a variety of modes of curvature formation, membrane localizations, and subtypes, characterization of a functional bacterial BAR domain protein has yet to be reported. Previously described membrane curvature-promoting proteins in bacteria are unrelated to BAR domain proteins or contain features ancillary to eukaryotic BAR domain protein function, such as amphipathic alpha helices (SpoVM, PmtA) (*Danne et al., 2017a*; *Danne et al., 2017b*; *Gill et al., 2015*; *Kim et al., 2017*), GTPase-associated signaling (FtsZ) (*Löwe and Amos, 1998*), or positive curvature localization (SpoVM, MamY) (*Gill et al., 2015*; *Kim et al., 2017*; *Toro-Nahuelpan et al., 2019*). However, actin-like (*van den Ent and Löwe, 2000*; *Pichoff and Lutkenhaus, 2005*), dynamin-like (*Low and Löwe, 2006*), and ESCRT-II-like (*Junglas, 2021*; *Liu, 2020*) proteins have been previously discovered in bacteria and archaea, suggesting that vesicle formation and tubule biogenesis can be achieved through evolutionarily conserved mechanisms.

*S. oneidensis* is a model organism for extracellular electron transfer (EET), a mode of respiration whereby metabolic electrons reduce exogenous terminal electron acceptors such as metals and electrodes. In *S. oneidensis*, this requires electrons to traverse the inner membrane, periplasm, and outer membrane via multiheme cytochromes that subsequently pass electrons to soluble electron shuttles (*Nealson and Scott, 2006*; *Marsili et al., 2008*) or directly to insoluble electron acceptors (*Kotloski and Gralnick, 2013*). The production of OMVs and OMEs in *S. oneidensis* is well documented, particularly upon surface attachment (*Chong et al., 2019*; *Subramanian et al., 2018*; *Pirbadian et al., 2014*; *Gorby et al., 2008*), and previous measurements showed that both are redox-active and can reduce extracellular iron, uranium, and technetium (*Gorby et al., 2008*; *Xu et al., 2018*). OMEs transition between membrane blebs or OMVs into chains of vesicles and tubules (*Pirbadian et al., 2014*); however, little is known about the mechanism controlling OME formation, shape, and curvature.

Here, we describe a component critical to the membrane morphology of *S. oneidensis* OMVs and OMEs as a putative BAR domain-like protein, which we term BdpA (BAR domain-like protein A). Through comparative proteomics, cryogenic electron microscopy, and molecular biology, we show that BdpA is enriched in OME/OMVs and influences both their diameter and shape. Likewise, expression

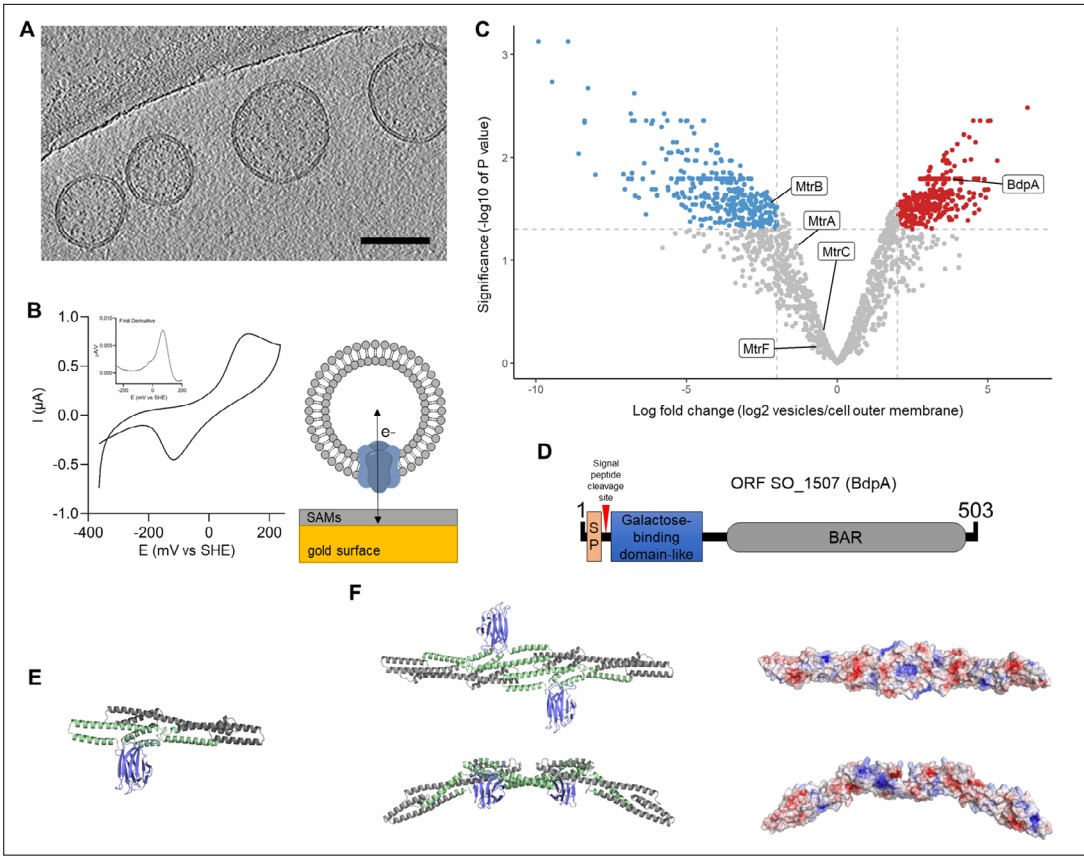

**Figure 1.** Redox active vesicles are enriched with Bin/Amphiphysin/RVS (BAR) domain-like protein BdpA. (**A**) Representative cryoelectron tomography image of *Shewanella oneidensis* MR-1 outer membrane vesicles (OMVs) (scale = 200 nm). (**B**) Cyclic voltammetry (scan rate of 10 mV/s) of vesicles adhered to gold electrode via small self-assembled monolayers, as diagrammed. Inset shows first derivative of anodic scan. (**C**) Volcano plot of vesicle proteome compared to cell-associated outer membrane (red = enriched in vesicles, blue = enriched in cell-associated outer membrane). Plot is representative of outer membrane vesicles and outer membrane fractions collected from three independent biological replicates each. (**D**) Schematic of putative BdpA domains. BdpA has a predicted signal peptide (SP) and cleavage site within the first 22–23 amino acid residues. (**E**) Ab initio predicted structure of the mature BdpA protein after signal peptide cleavage. Colors correspond to predicted domains in (**D**). (**F**) Top-down (top) and side profile (bottom) views of the predicted homodimeric structure of BdpA (left), and surface representation of the homodimeric protein without the galactose-binding domain-like region colored according to electrostatic potential (right), with positively charged residues in blue and negatively charged residues in red. The concave face (top right) has an accumulation of distributed positively charged residues, while the side profile shows predicted BAR domain-like intrinsic curvature.

The online version of this article includes the following figure supplement(s) for figure 1:

**Figure supplement 1.** Locations of proteins associated with vesicles and cell outer membrane predicted by PSORTb (CYT = cytoplasm, IM = inner membrane, OM = outer membrane, Peri = periplasm).

of BdpA elicited changes to membrane morphology in other bacteria, showing mechanistic evidence of BAR domain protein-mediated activity in vivo. This study frames future characterization efforts on other bacterial proteins with BAR domain-like activity in the context of OME/OMVs, and putative BAR domain-containing BdpA homologs in other bacteria suggests such domains may be widespread.

## Results and discussion
### Redox-active *S. oneidensis* OMVs are enriched with BdpA

OMVs were purified from *S. oneidensis* cells grown in batch cultures to characterize their redox features and unique proteome, as well as to identify putative membrane shaping proteins. Cryogenic

transmission electron microscopy (cryo-TEM) tomographic reconstruction slices of the purified samples used to assess membrane morphology showed uniform OMVs with the characteristic single membrane phenotype and an approximate diameter of 225 nm (*Figure 1A*). Cyclic voltammetry (CV) of OMVs adhered to a gold electrode via self-assembled monolayers was performed to assess redox activity of the samples (*Figure 1B*). First derivative analysis (*Figure 1B*, inset) revealed a prominent redox peak with a midpoint potential of 66 mV and a smaller peak at –25 mV vs. a standard hydrogen reference electrode (SHE), consistent with the characteristics of multiheme cytochromes such as MtrC/OmcA (*Xu et al., 2018*; *Okamoto et al., 2011*).

The proteome of OMVs was compared to the proteome of purified outer membranes extracted from whole cells. Using a label-free quantification method (*Sharma et al., 2015*), significant differences in the ratio of individual proteins in the vesicle to the outer membrane were determined (log fold change) (*Figure 1C*). Similar to other proteomics datasets (reviewed in *Nagakubo et al., 2019*), *S. oneidensis* vesicles contained proteins predicted to be localized to all subcellular fractions, for example, cytoplasm, inner membrane, periplasm, and outer membrane (*Figure 1—figure supplement 1*), and the total number of proteins detected were also comparable to more recent studies (*Yu et al., 2010*; *Lee et al., 2016*). The proteome of the purified OMVs showed 328 proteins were significantly enriched in the vesicles as compared to the outer membrane, and 314 proteins were significantly excluded from the vesicles (*Figure 1C*, *Supplementary file 1*). MtrCAB cytochromes were present in the OMVs as well as the outer membrane, supporting redox activity of OMVs observed by CV. Several proteins significantly enriched in the vesicles were identified that could contribute to OMV formation, including a putative murein transglycosylase (SO_2040), the peptidoglycan degradation enzyme holin (SO_2971), cell division coordinator CpoB (SO_2746), and a highly enriched putative BAR domain-containing protein encoded by the gene at open reading frame SO_1507, hereafter named BAR domain-like protein A (BdpA) (*Figure 1D*).

Vesicle enrichment of BdpA led us to hypothesize that BdpA could be involved in membrane shaping of OMVs based on the role of BAR domain proteins in eukaryotes. The C-terminal BAR domain of BdpA is predicted to span an alpha helical region from AA 276 to 451 (E-value = 2.96e-03); however, since the identification of the protein is based on homology to the eukaryotic BAR domain consensus sequence (cd07307), it is possible that the BAR domain region extends beyond these bounds (*Figure 1D*). Coiled-coil prediction (*Vincent et al., 2013*) suggests that BdpA exists in an oligomeric state of antiparallel alpha-helical dimers, as is the case for all known BAR domain proteins (*Frost et al., 2007*; *Linkner et al., 2014*; *Cui et al., 2013*; *Henne et al., 2007*). BdpA has an N-terminal signal peptide with predicted cleavage sites between amino acids 22 and 23, suggesting non-cytoplasmic localization (*Figure 1D*). A galactose-binding domain-like region positioned immediately downstream of the signal peptide supports lipid targeting activity seen in other BAR domain proteins, such as the eukaryotic sorting nexins (*van Weering and Cullen, 2014*). The *S. oneidensis* rough-type lipopolysaccharide (LPS) contains 2-acetamido-2-deoxy-D-galactose (*Vinogradov et al., 2003*), which could suggest localization of the protein to the outer leaflet of the outer membrane. Ab initio structure prediction generated without reliance on protein domain homology models produced through trRosetta (*Yang et al., 2020*) shows a C-terminal coiled-coil bundle of alpha helices corresponding to the predicted putative BAR domain-containing region, as well as an N-terminal jelly roll fold associated with galactose-binding domain-like domains (*Figure 1E*). Protein dimerization models were similarly generated using docking2 (*Lyskov and Gray, 2008*; *Chaudhury et al., 2011*; *Lyskov et al., 2013*), revealing an intrinsically curved dimer with positively charged residues along the concave surface (*Figure 1F*).

## BdpA controls size distribution of vesicles

In eukaryotic cells, BAR domain proteins are implicated in vesicle formation (*Daumke et al., 2014*; *Schöneberg et al., 2017*) and regulation of vesicle size (*Pinheiro et al., 2014*). To determine whether BdpA influences vesicle size in *S. oneidensis*, a *bdpA* deletion strain was constructed (Δ*bdpA*). A complement strain was also constructed in which the gene for BdpA was expressed in the mutant background (Δ*bdpA+ bdpA*) under control of the *phlF* promoter (p452-*bdpA*), which is inducible in *S. oneidensis* by addition of 2,4-diacetylphloroglucinol (DAPG) (*Yates et al., 2021*; *Meyer et al., 2019*). Growth curves for each strain show that the maximum optical density was lower for the Δ*bdpA+ bdpA* strain compared to the wild-type (WT) and Δ*bdpA* strains when grown with 1.25 or 12.5 μM DAPG

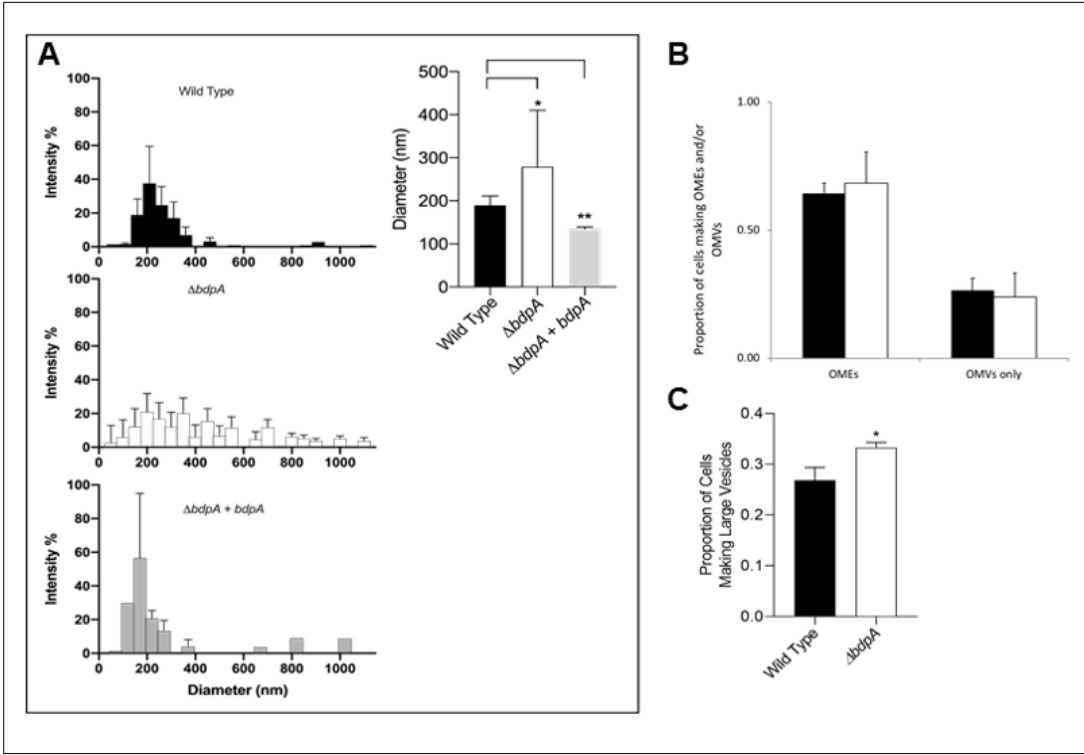

**Figure 2.** BdpA is responsible for maintaining vesicle size but does not alter the combined frequency of cells producing membrane structures. (**A**) Outer membrane vesicle (OMV) size distribution by dynamic light scattering (DLS) from the *Shewanella oneidensis* wild-type (WT) strain (top left, n = 11), deletion strain (Δ*bdpA*) (middle left, n = 9), and Δ*bdpA* strain expressing *bdpA* from a plasmid (bottom left, n = 3). Error bars represent the standard deviation from the average percentage of particles (% intensity) at a given diameter. The absence of an error bar indicates that a given diameter was only observed in a single replicate sample. The weighted average of the OMV diameter for each strain was also compared (right). Error bars represent the standard deviation and asterisks indicate a significant difference (p < 0.05, Student's t-test) between samples indicated by the line above. (**B**) Average proportion of cells producing distinctly visible outer membrane extensions (OMEs) (whether or not they also produce OMVs), and the average proportion of cells producing only visible OMVs (i.e., no distinctly visible OMEs) during 5 hr perfusion flow imaging experiments. Black bars are the WT strain and white bars are the Δ*bdpA* strain. A total of 2607 WT and 2943 Δ*bdpA* cells were quantified from n = 3 independent biological replicate experiments per strain. (**C**) Average proportion of cells forming large vesicles (typically >300 nm diameter) during 5 hr perfusion flow imaging experiments. A total of 1273 WT and 1317 Δ*bdpA* cells (from n = 3 independent biological replicate experiments per strain) were included in this quantification. Asterisk indicates a significant difference between samples (p < 0.0001, Pearson's $\chi^2$ test). For both (**B**) and (**C**). cell membranes were visualized by staining with FM 4–64 FX during time-lapse fluorescence imaging in a perfusion flow platform (6.25 ± 0.1 µL/s). Time-lapse images were acquired from at least five fields of view every 5 min over 5 hr for each strain. Error bars represent ± SEM.

The online version of this article includes the following figure supplement(s) for figure 2:

**Figure supplement 1.** Growth of *Shewanella oneidensis* strains in Luria Bertani (LB) (top) or *Shewanella* defined medium (SDM) (bottom) in response to DAPG exposure and BdpA induction.

**Figure supplement 2.** Anaerobic ferrihydrite reduction over time by *Shewanella oneidensis* strains.

**Figure supplement 3.** Example image of an outer membrane extension (OME) and a large vesicle produced by *Shewanella oneidensis* Δ*bdpA* after 3 hr during perfusion flow conditions. Scale = 2 µm.

(*Figure 2—figure supplement 1*). Ferrihydrite reduction assays were also performed to determine if EET was compromised in the *bdpA* deletion strain or induced complement. No difference was observed between WT or Δ*bdpA*, but Δ*bdpA+ bdpA* cells reduced ferrihydrite at a slightly delayed rate (*Figure 2—figure supplement 2*). These results suggest that if BdpA is involved in membrane remodeling, cell division could be compromised in the complement strain when expression of *bdpA* is not under native control, resulting in delayed growth. Growth and iron reduction were affected in

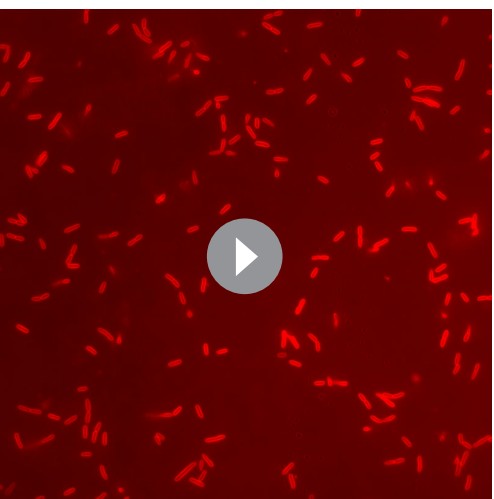

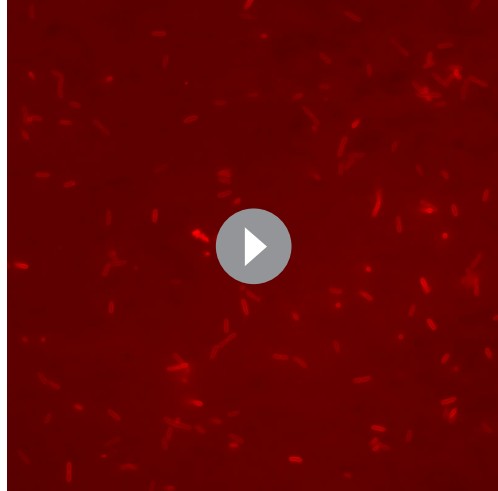

**Video 1.** Epifluorescence time course imaging of a single frame of *Shewanella oneidensis* wild-type (WT) cells during perfusion flow over a 5 hr duration, collecting images at 5 min intervals. Scale = 5 μm.
https://elifesciences.org/articles/60049/figures#video1

**Video 2.** Epifluorescence time course imaging of Δ*bdpA* cells during perfusion flow over a 5 hr duration, collecting images at 5 min intervals. Scale = 5 μm.
https://elifesciences.org/articles/60049/figures#video2

the Δ*bdpA+ bdpA* cultures even if no inducer was added, indicating leaky expression from the *phlF* promoter.

OMVs were harvested from all strains and their diameters were measured by dynamic light scattering (DLS). We observed a clear difference in the size distribution of OMVs between strains (*Figure 2A* left). WT OMVs had an average diameter of 190 nm with little variability in the population (standard deviation [s.d.] = ± 21 nm), while the diameters of Δ*bdpA* OMVs were distributed over a wider range with a significantly different average value of 280 nm (p = 0.0375, Student's t-test, s.d. = ± 131 nm) (*Figure 2A* right). The average diameter of OMVs from the Δ*bdpA+ bdpA* strain was 136 nm (s.d. = ± 4 nm), which was significantly different than WT (p = <0.001). The large difference in standard deviations between WT and Δ*bdpA* indicates a loss of control over the size of vesicles in the absence of *bdpA*. The same unpaired t-test with a Welch's post-correction does not detect a significant difference in the mean size of the vesicles between the WT and mutant strain. However, the F ratio to compare variances was a high 38.75, with a significant p-value of <0.00001. Similarly, the variances in vesicle size required the use of the Welch's post-correction to compare deletion strain and complement strain, which resulted in a significant difference between the two strains (p = 0.011). The variances were not significantly different between the WT and complement strain so the Welch's post-correction was not necessary. These results suggest that BdpA strongly influences vesicle diameter.

OMEs are known to transition between large vesicles and extensions over time (*Subramanian et al., 2018*; *Pirbadian et al., 2014*) and cell surface attachment influences OME formation (*Chong et al., 2019*). Therefore, cell-associated OME/OMV production and progression in the WT and Δ*bdpA* strains was measured in live cultures using a perfusion flow imaging platform previously demonstrated with *S. oneidensis* (*Subramanian et al., 2018*; *Pirbadian et al., 2014*). Cells attached to cover glass were counted and monitored for membrane features by automatically imaging at least five fields of view (~100–300 cells per field) for each biological replicate (n = 3) every 5 min for 5 hr to create a time-lapse series (examples provided in *Videos 1 and 2*). The average proportion of cells displaying either an OMV or OME relative to all cells was calculated for both strains. No significant difference was observed between the WT and Δ*bdpA* strains (*Figure 2B*). We hypothesized based on DLS results that the size of the OMVs, and possibly the shape of OMEs, would be different between strains. Fluorescence microscopy does not allow for the resolution required to visualize the size range of OMVs determined by DLS or differences in OME morphology. Therefore, we reanalyzed a subset of our data to quantify the proportion of cells that produced large vesicles where the outline of the stained lipid membrane (with a dark unstained interior) could clearly be distinguished within the limits of resolution of our measurements, typically those with an approximate diameter of 300 nm or greater (*Figure 2C*,

*Figure 2—figure supplement 3*). This included individual vesicles or those that were part of a vesicle chain or OME. The Δ*bdpA* strain had a significantly higher proportion of cells producing large vesicles than the WT (*Figure 2C*). Results of the perfusion flow experiment further substantiate DLS measurements despite the difference in growth condition (planktonic vs. surface-attached) indicating that BdpA influences vesicle size and supports the idea that BdpA has a key role in membrane architecture in *S. oneidensis* in vivo.

## BdpA constrains membrane extension morphology

Cryo-TEM was used to assess OMEs at the ultrastructural level in order to visualize morphological differences between OMEs of the WT, mutant, and complement strains. Cryo-TEM requires that cells be vitrified directly on EM grids. Therefore, the WT, Δ*bdpA, and* Δ*bdpA+ bdpA* strains were first visualized for OMV and OME frequency in static cultures using fluorescence microscopy to compare with results from perfusion flow microscopy. A portion of cells from overnight cultures (n = 3 biological replicates) were diluted and deposited onto a glass coverslip as previously described (*Chong et al., 2019*). Cells were imaged 3 hr post-deposition (five fields of view per replicate), and OMEs were observed for all strains (*Figure 3a*, *Videos 3–5*). Time-lapse imaging of representative fields of view for each strain over the course of 20 s highlights the motility that OMEs exhibit when formed in the absence of perfusion flow (*Videos 3–5*). Similar to perfusion flow experiments, no statistically significant difference in the overall frequency of OMEs was observed between strains (*Figure 3b*).

S. *oneidensis* OMEs from unfixed WT, Δ*bdpA,* and Δ*bdpA+ bdpA* strains were visualized by cryo-TEM at 90 min, a time point when initial OMEs are reliably observed, and 3 hr following cell deposition onto EM grids to assess if OME morphology changes over time. For all strains, OMEs were categorized as either tubules, narrow chains, or irregular chains (*Figure 3C and D*). Tubules were narrow OMEs with relatively uniform or slight symmetric curvature. Narrow chains were recorded as OMEs with a narrow, consistent diameter and symmetric curvature at constriction points. Irregular chains were classified as OMEs without a consistent diameter throughout the length of the OME and asymmetric curvature on either side of the extension. Blebs/bulges were outer membrane structures that did not resemble OMEs but still extended from the cell membrane surface and were also noted. At 90 min, OMEs from WT cells appeared as mostly narrow chains or tubule-like structures, and were seldom interspersed with lobed regions (*Figure 3E*, top), consistent with previous cryo-TEM analysis of *S. oneidensis* OMEs under similar conditions (*Subramanian et al., 2018*). OMEs from Δ*bdpA* cells had prevalent lobed regions with irregular curvature, but a few tubules were also observed (*Figure 3E*, middle). OMEs from the Δ*bdpA+ bdpA* complement strain were narrow tubules evenly interspersed with slight constriction points or 'junction densities' (*Subramanian et al., 2018*) extending from cells, along with prevalent bulging or blebbing of the outer membrane (*Figure 3E*, bottom). At 3 hr, the WT strain consistently displayed tubule OMEs (depicted in *Figure 3F*, top panel) or narrow chains of symmetric curvature. The Δ*bdpA* OMEs appear as lobed, disordered vesicle chains with irregular curvature compared to the WT, and vesicles can be observed branching laterally from lobes on the extensions (*Figure 3F*, middle panel). WT OMEs also exhibited lateral branching of vesicles and lobes, but OME curvature and diameter were qualitatively more uniform. Tubules were not observed in any Δ*bdpA* OMEs at 3 hr. OMEs from Δ*bdpA+ bdpA* cells appear as narrow tubules of a uniform curvature or as ordered vesicle chains similar to the WT strain (*Figure 3F*, bottom panel). The proportions of phenotypes varied substantially between strains suggesting that differences are due to the loss of *bdpA*. Qualitatively, this result is consistent with apparent loss of control of vesicle size revealed by DLS.

## Overexpression of BdpA results in OMEs during planktonic growth

Next, we sought to validate the effect of BdpA on membrane architecture observed with the mutant phenotype by introducing additional copies of the protein. In vitro tubule formation assays with purified proteins and liposomes are the canonical approach by which eukaryotic BAR domain proteins have been assessed for membrane sculpting activity (*Simunovic et al., 2015*). Localized BAR domain protein concentrations affect membrane shape, ranging from bulges to tubules and branched, reticular tubule networks at the highest protein densities (*Ayton et al., 2009*; *Simunovic et al., 2013*; *Noguchi, 2016*). However, molecular crowding of purified proteins with no documented membrane curvature formation activity, such as GFP, can also lead to ordering of liposomes into tubules (*Stachowiak et al.,*

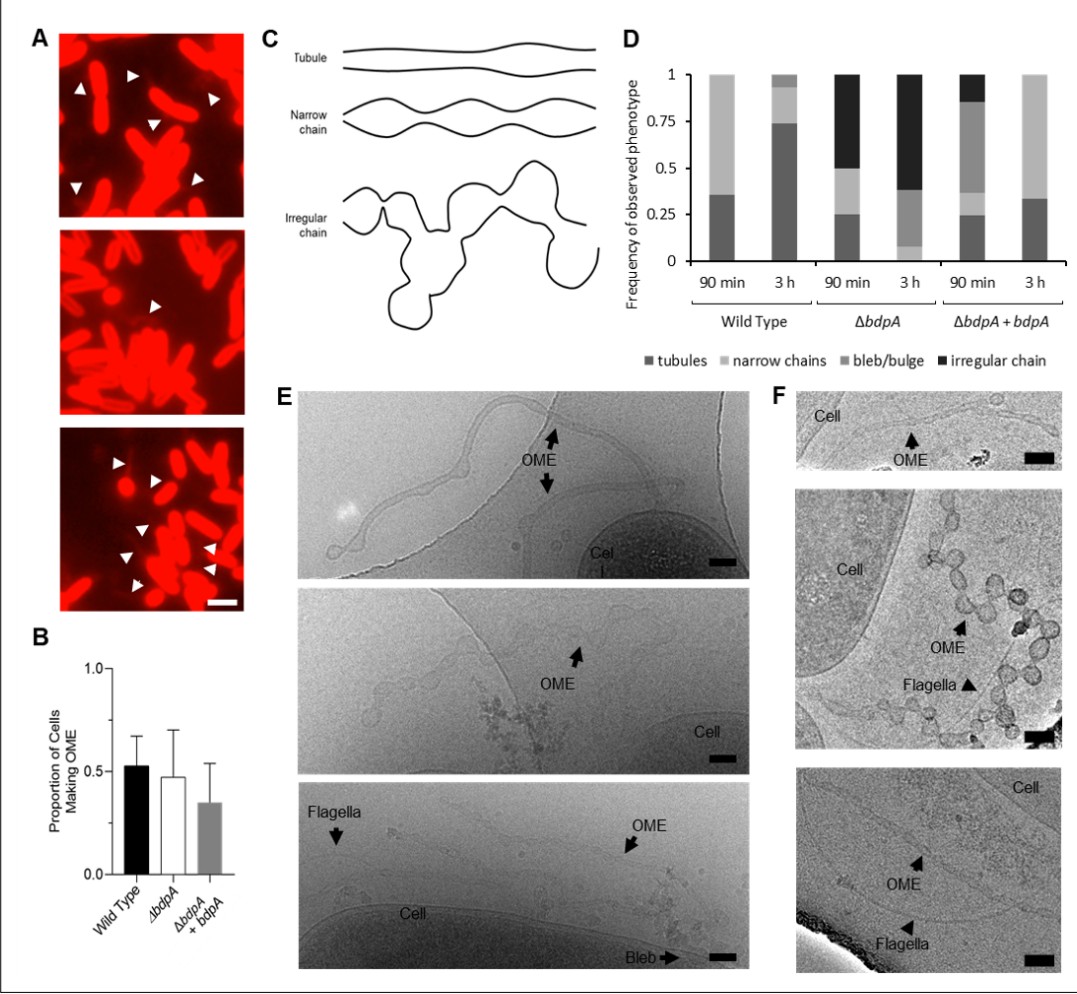

**Figure 3.** BdpA promotes outer membrane extension (OME) maturation into ordered tubules. (**A**) Fluorescence images of *Shewanella oneidensis* wild-type (WT) (top), Δ*bdpA* (middle), and Δ*bdpA+ bdpA* with 12.5 μM DAPG (bottom) OMEs. Scale = 2 μm. All cells were counted manually and categorized as either with extension or without extension (a total of 2444 cells from WT, 4378 cells from Δ*bdpA*, and 3354 cells from Δ*bdpA+ bdpA*). (**B**) Proportion of cells making OMEs relative to the total number of cells counted from static cultures at 3 hr post-deposition onto chambered cover glass, recorded from five random fields of view from fluorescence microscopy for each of three independent cultures per strain. Statistical significance was determined by Student's t-test (p > 0.05 for each). Error bars represent standard deviation. (**C**) Representative cartoon of OME phenotypes used for classification at 3 hr. (**D**) Frequency of OME phenotypes observed with cryogenic transmission electron microscopy (cryo-TEM) relative to the total number of OMEs observed from each strain. Phenotypes were documented from observations of 14 WT, 12 Δ*bdpA*, and 41 Δ*bdpA+ bdpA* OMEs at the 90 min time point, and 31 WT, 13 Δ*bdpA*, and 3 Δ*bdpA+ bdpA* OMEs at the 3 hr time point across three separate biological replicates, with two technical replicates of each strain per biological replicate. Membrane blebs/bulges were defined as non-structured membrane protrusions that did not resemble either of the other OME categories depicted in (**E**). (**E**) Representative cryo-TEM of *S. oneidensis* WT (top), Δ*bdpA* (middle), and Δ*bdpA+ bdpA* with 12.5 μM DAPG (bottom) OMEs at 90 min post-surface attachment. Scale = 100 nm. (**F**) Representative cryo-TEM images of WT (top), Δ*bdpA* (middle), and Δ*bdpA+ bdpA* with 12.5 μM DAPG (bottom) OMEs at 3 hr post-surface attachment. Scale = 100 nm.

*2012*). Further, tubule formation from liposomes is not limited to BAR domain protein activity and requires non-physiologically high protein concentrations (*Fröhlich et al., 2013*; *Ford et al., 2002*; *Yoon et al., 2010*). For these reasons, the effect of BdpA on membrane remodeling was tested by overexpression in the WT strain, as well as by orthogonal expression in two different host strains with no predicted BAR domain-containing proteins and no apparent OME production.

 *S. oneidensis* OMEs are more commonly observed in surface-attached cells than planktonic cells (*Chong et al., 2019*; *Subramanian et al., 2018*). BdpA was identified as expressed in planktonic

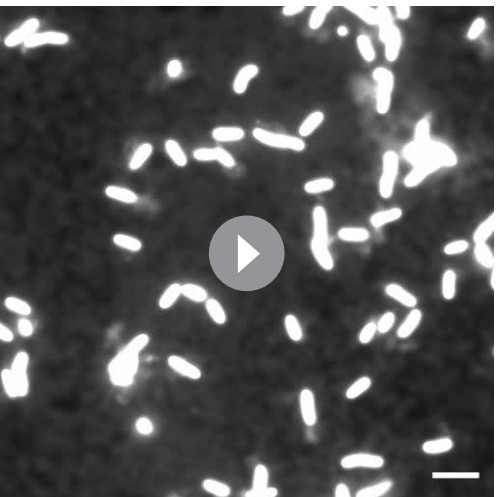

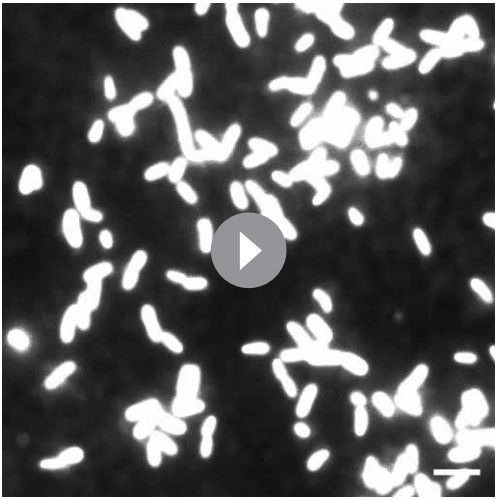

**Video 3.** Epifluorescence imaging of *Shewanella oneidensis* wild-type (WT) cells 3 hr post-deposition onto the surface of a chambered cover glass. Images were collected of a single field of view for a 20 s duration. Scale = 5 μm.

https://elifesciences.org/articles/60049/figures#video3

**Video 5.** Epifluorescence imaging of *Shewanella oneidensis* Δ*bdpA+ bdpA* cells 3 hr post-deposition onto the surface of a chambered cover glass. Images were collected of a single field of view for a 20 s duration. Scale = 5 μm.

https://elifesciences.org/articles/60049/figures#video5

cultures by proteomics, and its absence resulted in larger size vesicles during planktonic growth suggesting it may play a role in constraining membrane shape. We hypothesized that inducing expression of an additional copy of the *bdpA* gene in the WT strain during planktonic growth may further constrain the membrane, possibly resulting in OME formation prior to attachment. DAPG (12.5 μM) was added to planktonic cultures (n = 3 biological replicates) freshly inoculated from overnight cultures of the WT strain harboring p452-*bdpA* (WT+ *bdpA*) and incubated for 1 hr prior to deposition on cover glass for imaging. OMEs were observed at the outset of imaging indicating they had already formed during the incubation period (*Figure 4A*, left panels, *Video 6*). OMEs were not observed at this time point in the WT strain; however, OMEs were observed in the uninduced WT+ *bdpA* (not

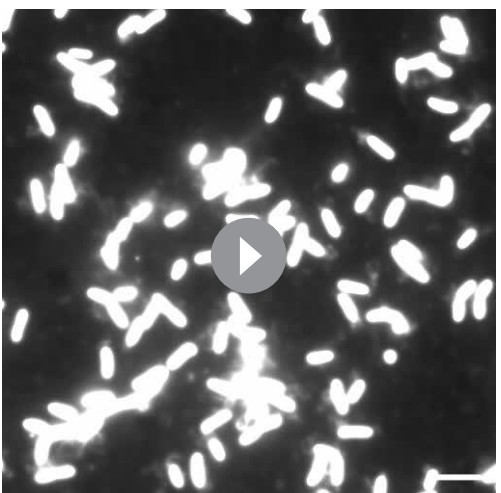

**Video 4.** Epifluorescence imaging of *Shewanella oneidensis* Δ*bdpA* cells 3 hr post-deposition onto the surface of a chambered cover glass. Images were collected of a single field of view for a 20 s duration. Scale = 5 μm.

https://elifesciences.org/articles/60049/figures#video4

shown), which as noted above for growth curves and iron reduction assays likely indicates leaky expression from the *phlF* promoter. Addition of DAPG or kanamycin did not induce OME formation in strains harboring the empty vector (not shown) indicating the observed OMEs were the result of BdpA expression.

The ultrastructure of OMEs from the WT + *bdpA* strain was examined by cryo-TEM 2 hr post-induction (n = 2 cultures). OMEs (n = 9 total OMEs observed) appeared as either tubules (6 out of 9 OMEs) or tubule-like segments interspersed with narrow vesicle chains proximal to the main cell body (3 out of 9 OMEs) (*Figure 4B*). Tubules and narrow chains observed in this mixed phenotype resembled those at 3 hr in surface-attached cells of the WT strain, confirming that the OME structures observed during planktonic induction are single-membrane periplasmic extensions and not cytoplasmic bacterial nanotubes (*Bhattacharya et al., 2019*; *Dubey et al., 2016*; *Dubey and Ben-Yehuda, 2011*; *Pande et al., 2015*; *Pospíšil et al., 2020*). Similar membrane

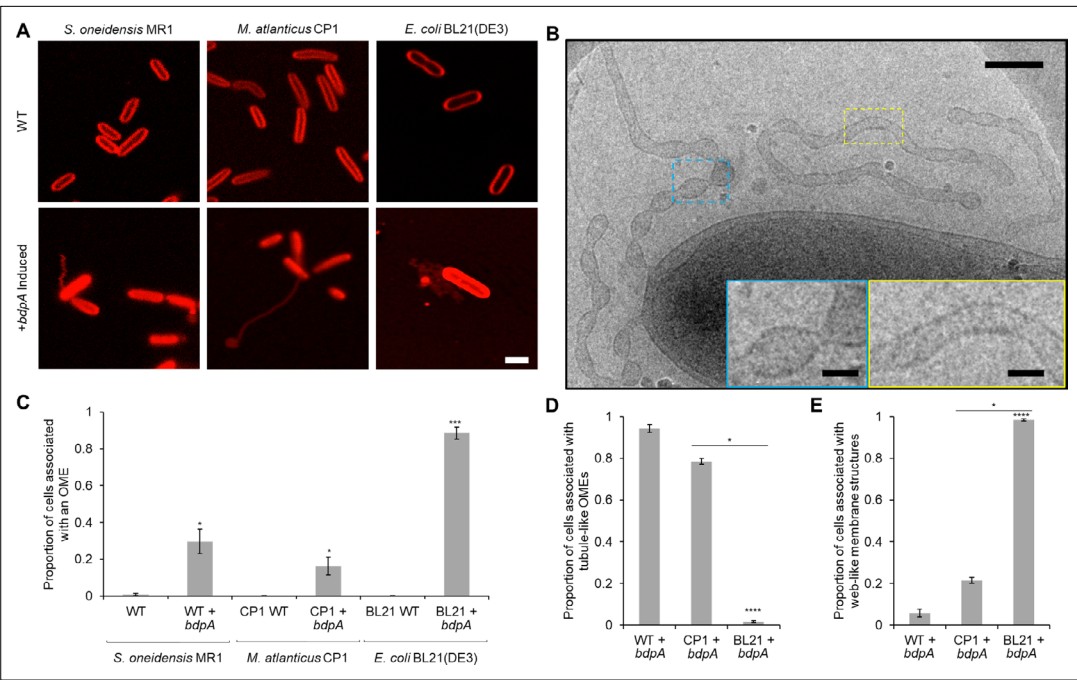

**Figure 4.** Heterologous expression of BdpA promotes outer membrane extension (OME) formation. (**A**) Induction (1 hr) of BdpA expression with 12.5 µM DAPG during planktonic, non-attached growth results in OME formation in *Shewanella oneidensis* (left, wild type [WT] + *bdpA*), *Marinobacter atlanticus* CP1 (middle, CP1+ *bdpA*), and *Escherichia coli* BL21(DE3) (right, BL21+ *bdpA*). Scale = 2 µm. At least three individual biological replicates were included per strain and images are representative of 5–15 fields of view per replicate. (**B**) Cryogenic transmission electron microscopy (cryo-TEM) image of OMEs following a 2 hr planktonic induction of BdpA expression in *S. oneidensis* WT + *bdpA* cells. Scale = 200 nm. Insets enlarged to show detail of regularly ordered electron densities at the surface of OME junctions (blue) and tubule regions (yellow). Scale = 50 nm. (**C**) Quantification of the proportion of cells associated with an OME from 1 hr planktonic induction cultures observed across 5–10 fields of view and three individual biological replicates from *S. oneidensis* WT (700 cells) and WT + *bdpA* (472 cells) (p = 0.025), *M. atlanticus* CP1 (4041 cells) and CP1+ *bdpA* (150 cells) (p = 0.041), and *E. coli* BL21 WT (2190 cells) and BL21+ *bdpA* (2623 cells) (p = 0.0007). Asterisks denote statistical significance between OME proportions of the WT and+ *bdpA* samples of the same species. No significance was observed between WT + *bdpA* and CP1+ *bdpA* (p = 0.089), but BL21+ *bdpA* produced more OMEs than either WT + *bdpA* (p = 0.0017) or CP1+ *bdpA* (p = 0.0001). (**D**) Proportion of the cells associated with a tubule-like OME relative to the total number of OME-associated cells observed for each+ *bdpA* strain. *S. oneidensis* WT + *bdpA* produced significantly more tubule-like OMEs than *E. coli* BL21+ *bdpA* cultures (p = 1.1 × 10⁻⁵) but not *M. atlanticus* CP1+ *bdpA* (p > 0.05). Similarly, more tubule-like OMEs were observed from *M. atlanticus* CP1+ *bdpA* cultures than in *E. coli* BL21+ *bdpA* (p = 0.035). (**E**) Proportion of the total number of OME-associated cells in each strain observed in contact with a web-like OME. *E. coli* BL21+ *bdpA* produced predominately web-like OMEs, and significantly more than *S. oneidensis* WT + *bdpA* (p = 1.1 × 10⁻⁵) or *M. atlanticus* CP1+ *bdpA* (p = 0.035). All statistical significance was determined by Welch's t-test. Error bars represent standard deviation.

The online version of this article includes the following figure supplement(s) for figure 4:

**Figure supplement 1.** Variability in outer membrane extension (OME) phenotypes following BdpA induction in *Marinobacter atlanticus* CP1+ *bdpA* cells. Cells displayed an array of membrane curvature phenotypes, ranging from tubule-like OMEs (**A–F**), membrane vesicles or blebbing (**C,F**), and branched, web-like OME/outer membrane vesicle (OMV) chains (**F**). Frequencies of each phenotype are shown in *Figure 4*. Scale = 2 µm.

---

sculpting phenotypes showing a mixture of tubules and pearled, 'beads on a string' segments within the same field of view were observed previously from in vitro cryo-TEM experiments using liposomes and purified F-BAR protein Pacsin1 from eukaryotic cells (*Wang et al., 2009*). Thus, as predicted, BdpA may constrict or scaffold OMVs produced during planktonic growth into OMEs (*Chong et al., 2019*). Additional biochemical and biophysical assays beyond the scope of this initial manuscript are needed to further elucidate the effect of BdpA on membrane sculpting.

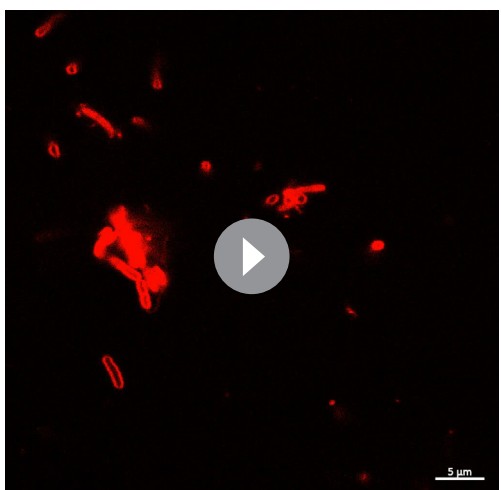

**Video 6.** Confocal imaging of *Shewanella oneidensis* MR-1 p452-*bdpA* cells after 1 hr planktonic induction of BdpA with 12.5 μM DAPG. Cells with apparent outer membrane extensions (OMEs) can be seen moving through the field of view. Scale = 5 μm.
https://elifesciences.org/articles/60049/figures#video6

## BdpA-mediated membrane extensions in *Marinobacter atlanticus* and *E. coli*

In order to test the effect of BdpA on membrane architecture in bacterial species with no identified BdpA homolog, the p452-*bdpA* expression vector was transformed into *M. atlanticus* (*Bird et al., 2018*) and *E. coli*. *Marinobacter* and *Shewanella* are of the same phylogenetic order (Alteromonadales), and *M. atlanticus* has been used for expression of other *S. oneidensis* proteins (*Bird et al., 2019*). *E. coli* strain BL21(DE3) is optimized for protein overexpression. Cultures (n = 3 biological replicates for each strain) of WT and transformed *M. atlanticus* and *E. coli* BL21(DE3) were prepared in an identical manner to those used for *S. oneidensis* planktonic expression experiments described above. OMEs were observed in both strains harboring the p452-*bdpA* vector following induction of *bdpA* expression by DAPG (12.5 μM) for 1 hr prior to deposition on cover glass, and representative images are shown in *Figure 4A* (middle and right panels). OMEs were observed from cells in all imaged fields (5–10 fields of view for each replicate). *M. atlanticus* CP1+ *bdpA* OMEs ranged from small membrane blebs to tubules extending beyond 10 μm in length from the surface of the cell (*Figure 4—figure supplement 1*). *E. coli* BL21(DE3)+ *bdpA* cells (*Figure 4A*, right panels) also displayed OMEs; however, in most cases OMEs appeared as a web-like network of reticular membrane structures. WT cells for each strain had uniform, continuous cell membranes. OMEs were significantly more prevalent in all cells with the *bdpA* plasmid than WT cells (*Figure 4C*). The *E. coli* BL21(DE3)+ *bdpA* strain had the most cells associated with outer membrane structures per total number of cells per field. This result was expected, since this *E. coli* strain was designed for protein overexpression. *S. oneidensis* WT + *bdpA* and *M. atlanticus* CP1+ *bdpA* displayed more tubule-like OMEs than *E. coli* BL21(DE3)+ *bdpA* (p = 1.1 × 10$^{-5}$ and p = 0.035, respectively) (*Figure 4D*). The majority of OMEs associated with *E. coli* BL21(DE3)+ *bdpA* cells had the web-like, reticular morphology noted in *Figure 4A*, right panels (quantified in *Figure 4E*). These results show that expression of *S. oneidensis* BdpA results in membrane remodeling into structures resembling those associated with the presence of BAR domain-containing proteins from in vitro eukaryotic BAR protein experiments (*Ayton et al., 2009*; *Simunovic et al., 2013*; *Noguchi, 2016*).

## Phylogenetic analyses support mechanistic and structural data for BdpA as a bacterial BAR domain-like protein

The discovery of a novel membrane sculpting protein with BAR domain-like activity and predicted structural similarity in bacteria provokes questions into the evolutionary origin of BAR domains, such as whether the putative BdpA BAR-like domain in *S. oneidensis* arose as a result of convergent evolution, a horizontal gene transfer event, or has a last common ancestor across all domains of life. A PSI-BLAST (*Altschul et al., 1997*) search against the NCBI nr database was performed using the BAR domain amino acid sequence of BdpA as the initial query to identify BdpA homologs in other organisms. BdpA homologs were annotated as hypothetical proteins in all of the species identified. In the initial round, 24 proteins were found from other organisms identified as *Shewanella* with a high conservation among the proteins and another 28 proteins were found in more distant bacterial species that had an amino acid identity of 65–44%. A second iteration identified a few proteins from more distantly related bacterial species, followed by proteins from the eukaryotic phylum *Arthropoda* that were annotated as being centrosomal proteins. Only five of the proteins from the search returned hits to the position-specific scoring matrix (PSSM) of the BAR cd07307, but two did not have any unique residues within the BAR domain. Overall, BdpA barely met the criteria to be assigned as matching

the BAR domain via PSSM models. The rest of the protein homologs identified had enough differences to fail to match the BAR model despite all having greater than 44% amino acid identity to the *S. oneidensis* BdpA sequence. An attempt was made to build a hidden Markov model (HMM) using hmmer (*Finn et al., 2011*) to use for searching for other proteins that might match, but as with the PSI-BLAST search, only the proteins that formed the model were returned as good matches. This indicates that while sequence homology between BdpA and the existing BAR domain consensus sequence predicted the BAR domain region in BdpA using hmmer or NCBI tools, the sequence conservation is at the cusp of a positive hit by the HMM since other closely related (>90% homology) BdpA orthologs were not predicted to contain a BAR domain by this method. The eukaryotic protein that is most similar to BdpA (27 % amino acid identity) is a putative centrosomal protein in the ant *Vollenhovia emeryi* (accession #: XP_011868153) that is predicted to contain an amino terminal C2 membrane-binding domain and a carboxy-terminal SMC domain within a coiled-coil region. Despite CDD search failing to predict the presence of a BAR domain in this protein, it does not preclude the presence of one, pending an updated BAR Pfam HMM. Alternatively, the homology could be due to convergent evolution because of the necessary spacing of polar amino acids to maintain the coiled-coil structure in both BdpA and XP_011868153.

Fifty-two BdpA homologs were identified by PSI-BLAST in most but not all species of *Shewanella*, as well as *Alishewanella*, *Rheinheimera*, and *Cellvibrio*. The current BAR domain Pfam HMM prediction analysis identified BAR domain features in only 5 of the 52 bacterial homologs despite greater than 90 % homology to *S. oneidensis* BdpA. An amino acid alignment of the 52 BdpA homologs was used to generate a maximum likelihood phylogenetic tree showing evolutionary relatedness of BdpA homologs to the 23 eukaryotic BAR domains that were used to build the Pfam HMM (*Figure 5—figure supplement 1*). This shows that the BdpA homologs identified by PSI-BLAST form a cohesive group with most sequences forming clades by genus. The four unique BAR domain sequences from the five BdpA homologs predicted to contain a BAR domain based on the current model were subsequently aligned with representative known BAR domain-containing proteins from the various BAR domain subtypes (N-BAR, F-BAR, and I-BAR) (*Salzer et al., 2017*), bacterial proteins with known membrane remodeling phenotypes (*Tanaka et al., 2010*; *Danne et al., 2017a*; *Löwe and Amos, 1998*), along with the most diverse (*Lu et al., 2020*) BAR domain CDD sequences from eukaryotes (*Supplementary file 2*). In this maximum likelihood phylogenetic tree generated from this alignment, the BdpA sequences form a cohesive clade separate from other bacterial proteins that associate with membranes or influence curvature in vitro and instead clusters more closely with eukaryotic BAR domain-containing proteins (*Figure 5*). Bacterial proteins (PmtA, MamY, and FtsA) unrelated to BdpA or any of the other putative BAR domain-containing proteins are scattered elsewhere in the tree, suggesting that there is no direct evolutionary connection between them.

The maximum likelihood phylogenetic tree built from this alignment shows that BdpA and its homologs have branch lengths shorter than those between eukaryotic BAR domain-containing proteins, suggesting that BdpA likely contains a functional, yet unique BAR domain. Induction of BdpA expression in the Δ*bdpA+ bdpA* strain displayed attenuated growth (*Figure 2—figure supplement 1*). It is possible that BdpA could have redundant activity with FtsA, another alpha-helical protein known to stabilize FtsZ during contractile ring formation and cell division (*Pichoff and Lutkenhaus, 2005*). Crowding of BdpA at the membrane could potentially inhibit the ability of FtsA to interact with FtsZ. Further investigation into the structure and function of BdpA is needed to determine its role in cell division. Likewise, structural and in vitro biophysical data are required before confirming that this mechanistically similar protein represents a new class of bacterial BAR domain-containing proteins.

The evolutionary origin of this clade of protein remains unclear from the sequences currently available. It is possible that BdpA arose as a result of convergent evolution due to selection of positively charged amino acids at key locations along the helices. There is also a possibility that the predicted BAR domain-containing region arose as a result of horizontal gene transfer from a eukaryote due to the prevalence of eukaryotic coiled-coil proteins with predicted homology to BdpA after two iterations of PSI-BLAST. However, the branch lengths and low bootstrap values supporting the placement of many of the BAR domain subtypes prevent us from directly inferring the evolutionary history of BAR domains. Discovery of other putative bacterial BAR proteins would help to build this analysis, and if future comparative proteomics analysis of OME/OMVs demonstrates overlapping activity of BdpA with preferential cargo loading into OME/OMVs, it could hint at the evolutionary origins of

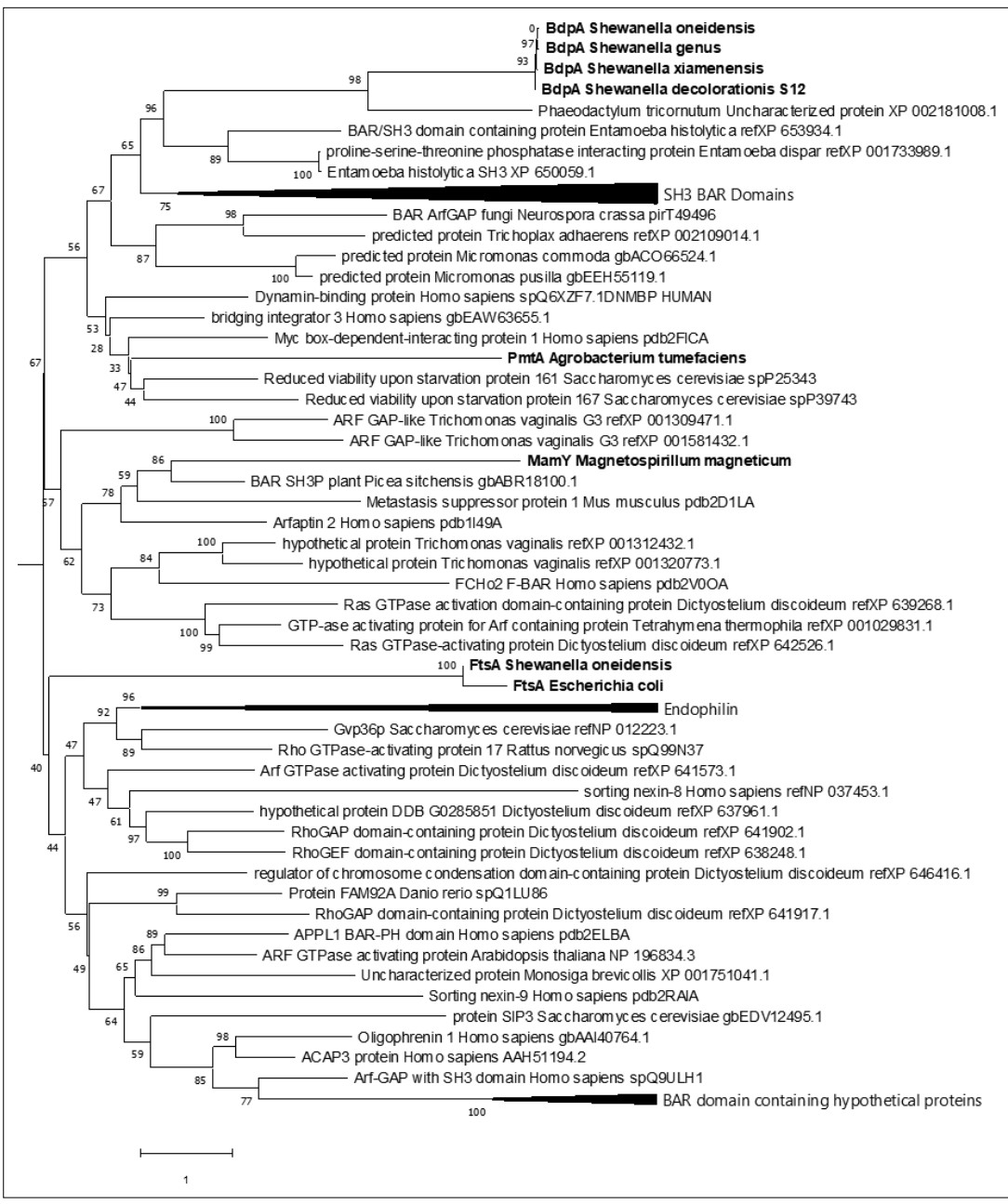

**Figure 5.** Comparative phylogenetic analysis of BdpA with bacterial homologs, membrane curvature-associated bacterial proteins, and eukaryotic Bin/Amphiphysin/RVS (BAR) domains. Maximum likelihood evolutionary histories were inferred from 1000 bootstrap replicates, and the percentage of trees in which the taxa clustered together is shown next to the branches. Arrows indicate multiple branches collapsed to a single node. *Shewanella oneidensis* BdpA and three unique bacterial orthologs (WP_011623497 – unclassified *Shewanella*, ESE40074 – *S. decolorationis* S12, KEK29176 – *Streptomyces xiamenensis*) predicted by the current BAR domain Pfam hidden Markov model (HMM) to contain a BAR domain aligned with representative BAR domains from various BAR domain subtypes (N-BAR, F-BAR, SNX-BAR, I-BAR) at a total of 435 positions. Bacterial proteins are emphasized with bold font. The gamma distribution used to model evolutionary rate differences among sites was 7.24.

The online version of this article includes the following figure supplement(s) for figure 5:

**Figure supplement 1.** BdpA has homologs in other bacterial species.

vesicle-based protein trafficking. Conservation of BAR domain proteins supports the notion that three-dimensional organization of proteins in lipid structures is as important to bacteria as it is eukaryotes, and suggests that additional novel bacterial BAR domain-like proteins are waiting to be discovered.

## Conclusion

*S. oneidensis* expresses a functional membrane sculpting protein with BAR domain-like activity and homology, which is the first identified and characterized in bacteria. Enrichment of BdpA in the redox-active OMVs and BdpA-mediated sculpting of the OMVs into a uniform diameter suggests over-lapping mechanistic functionality with eukaryotic BAR proteins in the context of vesicle constriction (*Daumke et al., 2014*). This finding was further demonstrated through fluorescence microscopy during perfusion flow, where more cells produced large vesicles in the absence of BdpA. Membrane constric-tion activity of BdpA was confirmed through cryo-EM images that depicted disordered tubules in the absence of BdpA. Variation in OMEs with BdpA ranged from ordered, narrow vesicle chains of a consistent diameter to stable tubules. The closest phylogenetic eukaryotic BAR domain subtype to BdpA, F-BAR domains, exhibits similar variation in tubule morphology, depending upon the orienta-tion of the tip-to-tip oligomerization around the tubules (*Frost et al., 2007*; *Shimada et al., 2007*; *Wang et al., 2009*). While the presence of the predicted galactose-binding domain-like region at the N-terminal end of the protein suggests possible localization to the outer membrane, we cannot rule out that membrane sculpting phenotypes observed with BdpA are an indirect consequence of binding to LPS or through association with periplasmic components, such as peptidoglycan, at this time. Subsequent studies will include vesicle constriction into tubules in vitro with purified protein to ascertain the extent of functional mechanistic similarity of BdpA to other F-BAR proteins. Ulti-mately, the discovery of BdpA and its homologs presents a critical step in understanding the impact of membrane sculpting proteins with BAR domain-like activity on extracellular membrane structures.

## Materials and methods

**Key resources table**

| Reagent type (species) or resource | Designation | Source or reference | Identifiers | Additional information |
|---|---|---|---|---|
| Gene (*Shewanella oneidensis*) | *bdpA* | GenBank | AE014299.2 | locus tag SO_1507 |
| Strain, strain background (*S. oneidensis* MR-1) | WT | *Myers and Nealson, 1988* | | Wild type |
| Strain, strain background (*S. oneidensis* MR-1) | WT+ pBBR1-mcs2 | This paper | | Wild type with the pBBR1-mcs2 empty vector |
| Strain, strain background (*S. oneidensis* MR-1) | WT + *bdpA* | This paper | | Wild type with an extra copy of *bdpA* in trans under inducible control by DAPG on the p452-bdpA plasmid |
| Strain, strain background (*S. oneidensis* MR-1) | Δ*bdpA* | This paper | | *bdpA* scarless deletion |
| Strain, strain background (*S. oneidensis* MR-1) | ΔbdpA + pBBR1-mcs2 | This paper | | *bdpA* knockout strain with the pBBR1-mcs2 empty vector |
| Strain, strain background (*S. oneidensis* MR-1) | Δ*bdpA*+ *bdpA* | This paper | | *bdpA* scarless deletion with *bdpA* under inducible control by DAPG in the p452-bdpA plasmid |
| Strain, strain background (*S. oneidensis* MR-1) | JG1194 (ΔMtr) | *Coursolle and Gralnick, 2010* | | *S. oneidensis* with the extracellular electron transfer pathway proteins deleted (Δ*mtrC*/Δ*omcA*/Δ*mtrF*/Δ*mtrA*/Δ*mtrD*/Δ*dmsE*/Δ*SO4360*/Δ*cctA*/Δ*recA*) |

*Continued on next page*

*Continued*

| Reagent type (species) or resource | Designation | Source or reference | Identifiers | Additional information |
|---|---|---|---|---|
| Strain, strain background (*S. oneidensis* MR-1) | JG1194 (ΔMtr)+ pBBR1-mcs2 | This paper | | White strain harboring the pBBR1-mcs2 empty vector |
| Strain, strain background (*Marinobacter atlanticus* CP1) | CP1 | *Bird et al., 2018* | | Wild type |
| Strain, strain background (*M. atlanticus* CP1) | CP1+ *bdpA* | This paper | | Heterologous expression strain of *bdpA* under inducible control by DAPG from the p452-*bdpA* plasmid in *M. atlanticus* CP1 |
| Strain, strain background (*Escherichia coli*) | BL21(DE3) | PMID:3537305 | | OneShot *E. coli* BL21(DE3) |
| Strain, strain background (*E. coli*) | BL21+ *bdpA* | This paper | | *E. coli* BL21(DE3) with *bdpA* under inducible control by DAPG in the p452-bdpA plasmid |
| Strain, strain background (*E. coli*) | UQ950 | *Saltikov and Newman, 2003* | | Cloning strain |
| Strain, strain background (*E. coli*) | BW29427 (WM3064) | *Saltikov and Newman, 2003* | | Conjugation strain |
| Recombinant DNA reagent | pBBR1-mcs2 (plasmid) | *Kovach et al., 1995* | | Empty vector |
| Recombinant DNA reagent | pBBJM (plasmid) | This paper | | Cloning backbone |
| Recombinant DNA reagent | pSMV3 (plasmid) | *Simon et al., 1983* | | Suicide vector |
| Recombinant DNA reagent | pBBJM-452 (plasmid) | *Yates et al., 2021*; *Meyer et al., 2019* | | Marionette sensor with yellow fluorescent protein (YFP) under inducible control of DAPG in the pBBR1-mcs2 backbone |
| Recombinant DNA reagent | pSMV3_1507KO (plasmid) | This paper | | Contains up- and downstream regions of open reading frame SO_1507 (*bdpA*) |
| Recombinant DNA reagent | p452-*bdpA* (plasmid) | This paper | | DAPG inducible *bdpA* in the pBBJM-452 plasmid instead of YFP |
| Sequence-based reagent | pAJMF2 | This paper | PCR primers | TTAACGCGAATTTTAACAAAATATTAACGC cccgcttaacgatcgttggctg |
| Sequence-based reagent | pAJMR3 | This paper | PCR primers | AGCGGATAACAATTTCACACAGGAAACAGC Tacctcagataaaatatttgc |
| Sequence-based reagent | pBBRF3 | This paper | PCR primers | gggctcatgagcaaatattttatctgaggt AGCTGTTTCCTGTGTGAAATTG |
| Sequence-based reagent | pBBRR2 | This paper | PCR primers | acccgcgctcagccaacgatcgttaagcggg GCGTTAATATTTTGTTAAAATTCGC |
| Sequence-based reagent | 1507 F_insert | This paper | PCR primers | ttaatactagagaaagaggggaaatactag ATGCGCACCGCTGC |
| Sequence-based reagent | 1507 R_insert | This paper | PCR primers | gaggcctcttttctggaatttggtaccgagC TACATAAAGGCTTTAGTAAAGGCTT |
| Sequence-based reagent | BBJMV_reverse | This paper | PCR primers | CAGCATTGAGATGACTGCAGCGGTGCGCAT ctagtatttcccctctttctctagtat |
| Sequence-based reagent | BBJMV_forward | This paper | PCR primers | AAGGAAGCCTTTACTAAAGCCTTTATGTAG ctcggtaccaaattccagaaaag |
| Sequence-based reagent | pSMV3_R | This paper | PCR primers | GCTAATCCAAAGGGAAACACCACA ATAAACGATCCCCCGGGCTG |

*Continued on next page*

*Continued*

| Reagent type (species) or resource | Designation | Source or reference | Identifiers | Additional information |
| --- | --- | --- | --- | --- |
| Sequence-based reagent | pSMV3_F | This paper | PCR primers | Caagacattattgaaattaagcaaagcacacactagttctagagcggccg |
| Sequence-based reagent | bdpAUpstream1kb_F | This paper | PCR primers | tgatatcgaattcctgcagcccgggggatcgtttattgtggtgtttccctttgga |
| Sequence-based reagent | bdpAUpstream1kb_R | This paper | PCR primers | AAGCCCAGTAAACCTTTCTATAACAAGTCGAAAAGCCT CATAAAACATAAATAACATACGAAG |
| Sequence-based reagent | bdpAdwnstream1kb_F | This paper | PCR primers | cgtatgttatttatgtttttatgaggcttttcgacttgttatagaaaggtttactggg |
| Sequence-based reagent | bdpAdwnstream1kb_R | This paper | PCR primers | ACCGCGGTGGCGGCCGCTCTAGAACTAGTGTGTGC TTTGCTTAATTTCAATAATGTCTTG |
| Other | FM 4–64 | Invitrogen | T13320 | (0.25 µg/mL) |

## Bacterial strains, plasmids, and medium

The bacterial strains used in this study can be found in the Key resources table. *S. oneidensis* strains were grown aerobically in Luria Bertani (LB) media at 30 °C with 50 µg/mL kanamycin when maintaining the plasmid. *E. coli* strains were likewise grown in LB with 50 µg/mL kanamycin, but at 37 °C. For microscopy experiments with *S. oneidensis* strains, cells were centrifuged and resuspended in the *Shewanella* defined media (SDM) comprised of 30 mM PIPES, 60 mM sodium DL-lactate as an electron donor, 28 mM $NH_4Cl$, 1.34 mM KCl, 4.35 mM $NaH_2PO_4$, 7.5 mM NaOH, 30 mM NaCl, 1 mM $MgCl_2$, 1 mM $CaCl_2$, and 0.05 mM ferric nitrilotriacetic acid (*Pirbadian et al., 2014*). *M. atlanticus* CP1 strains were grown in BB media (50 % LB media, 50 % Marine broth) at 30 °C with 100 µg/mL kanamycin to maintain the plasmids as described previously (*Bird et al., 2018*).

Inducible BdpA expression plasmids were constructed for use in *S. oneidensis* MR-1, *M. atlanticus* CP1, and *E. coli* BL21(DE3) using the pBBR1-mcs2 backbone described previously (*Bird et al., 2018*). The Marionette sensor components (*phlF* promoter, constitutively expressed PhlF repressor, and yellow fluorescence protein [YFP]) cassette from pAJM.452 (*Meyer et al., 2019*) was cloned into the pBBR1-mcs2 backbone, and the YFP cassette was replaced with the full-length gene (1511 nucleotides) encoding BdpA by Gibson assembly (primers in the Key resources table). The resulting plasmid was given the name p452-*bdpA*. The Gibson assembly reactions were electroporated into *E. coli* Top10 DH5α cells (Invitrogen), and the sequences were confirmed through Sanger sequencing (Eurofins genomics). Plasmid constructs were chemically transformed into conjugation-competent *E. coli* WM3064 cells for conjugative transfer into the recipient bacterial strains of *S. oneidensis* MR-1 and *M. atlanticus* CP1. The same BdpA expression vector was transformed into *E. coli* BL21(DE3) cells (Invitrogen) by chemical transformation.

Generation of a scarless Δ*bdpA* knockout mutant of *S. oneidensis* was performed by combining 1 kilobase fragments flanking upstream and downstream from *bdpA* by Gibson assembly into the pSMV3 suicide vector. The resultant plasmid pSMV3_1507KO was transformed into *E. coli* DH5α λ *pir* strain UQ950 cells for propagation. Plasmid sequences were confirmed by Sanger sequencing before chemical transformation into *E. coli* WM3064 for conjugation into *S. oneidensis*. Conjugation of pSMV3_1507KO into *S. oneidensis* MR-1 was performed as described previously (*Gorby et al., 2006*). The empty vector pBBR1-mcs2 was transformed into *S. oneidensis* WT, Δ*bdpA*, and strain JG1194 lacking the Mtr pathway proteins for ferrihydrite reduction and growth curve analysis.

Optical densities at 600 nm were measured to determine growth curves for each strain in either LB or SDM with 50 µg/mL kanamycin and DAPG as indicated. Cultures of 500 µL samples diluted to an initial $OD_{600}$ of 0.1 were grown in Costar polystyrene 48-well plates (Corning Incorporated) incubated within a Tecan Infinite M1000 Pro (Grödig, Austria) plate reader at 30 °C with shaking agitation at 258 rpm. Optical densities were recorded every 15 min with i-Control software (2.7). All measurements were performed on three independent biological replicates per strain.

## Purification of OMVs

*S. oneidensis* MR-1 cells were grown in LB in 1 L non-baffled flasks at 30 °C at 200 rpm. When an $OD_{600}$ of 3.0 was reached, cells were pelleted by centrifugation at 5000 × *g* for 20 min at 4 °C, and resulting supernatant was filtered through a 0.45 µm filter to remove remaining bacterial cells. Vesicles were obtained by centrifugation of the filtered supernatant at 38,400 × *g* for 1 hr at 4 °C in an Avanti J-20XP centrifuge (Beckman Coulter, Inc). Pelleted vesicles were resuspended in 20 mL of 50 mM HEPES (pH 6.8) and filtered through 0.22 µm pore size filters. Vesicles were again pelleted as described above and finally resuspended in 50 mM HEPES, pH 6.8, except for vesicle preparations used for electrochemistry which were suspended in 100 mM MES, 100 mM KCl, pH 6.8. This more inclusive purification method was chosen to prevent biasing the results in favor of vesicles of specific sizes and increase the likelihood of isolating more structures, which might be a concern with density gradient centrifugation. As a result, co-purification of extracellular DNA, flagella, and pili was expected. This protocol was adapted from *Pérez-Cruz et al., 2013*.

## Cryoelectron tomography

Vesicle samples were diluted to a protein concentration of 0.4 mg/mL and applied to glow-discharged, carbon-coated, R2/2, 200 mesh copper Quantifoil grid (Quantifoil Micro Tools) using a Vitrobot chamber (FEI). Grids were automatically plunge-frozen and saved for subsequent imaging. No fixative was used. Images were collected on an FEI Krios transmission electron microscope equipped with a K2 Summit counting electron-detector camera (Gatan). Data were collected using customized scripts in SerialEM (*Hagen et al., 2017*), with each tilt series ranging from −60° to 60° in 3° increments, an underfocus of ~1–5 µm, and a cumulative electron dose of 121 $e/A^2$ for each individual tilt series. Tomograms were reconstructed using a combination of ctffind4 (*Rohou and Grigorieff, 2015*) and the IMOD software package (*Kremer et al., 1996*).

## Electrochemistry

CHA Industries Mark 40 e-beam and thermal evaporator was used to deposit a 5 nm Ti adhesion layer and then a 100 nm Au layer onto cleaned glass coverslips (43 mm × 50 mm #1 Thermo Scientific Gold Seal Cover Glass, Portsmouth NH). Self-assembled monolayers were formed by incubating the gold coverslips in a solution of 1 mM 6-mercaptohexanoic acid in 200 proof ethanol for at least 2 hr. The electrodes were then rinsed several times in ethanol followed by several rinses in MilliQ water. The SAMs layer was then activated by incubation in 100 mM *N*-(3-imethylaminopropyl)-*N'*-ethylcarbodiimide hydrochloride and 25 mM *N*-hydroxysuccinimide, pH 4, for 30 min. A sample of OMVs was deposited on the surface of the electrode and incubated at room temperature overnight in a humid environment. CV, at a scan rate of 10 mV/s from −400 to +200 mV vs. SHE, was performed in a 50 mL three-electrode half-cell completed with a platinum counter electrode, and a 1 M KCl Ag/AgCl reference electrode electronically controlled by a Gamry 600 potentiostat (Gamry, Warminster, PA). The whole experiment was completed in an anaerobic chamber with 95 % nitrogen, 5 % hydrogen atmosphere.

## Proteomics

Vesicle samples were prepared for proteomics analysis as described above and a portion of the proteomics samples were used for DLS measurements. Different cultures on different days were used for outer membrane extraction. The same culture conditions used for vesicle harvesting were used to grow cells for outer membrane extraction, with the exception of culture flask volume due to differences in biomass required. The outer membrane (OM) fraction was purified via the Sarkosyl method described by *Brown et al., 2010*. Three independent 50 mL overnight cultures of cells were harvested by centrifugation at 10,000× *g* for 10 min. The cell pellets were resuspended in 20 mL of 20 mM ice-cold sodium phosphate (pH 7.5) and passed four times through a French Press (12,000 $lb/in^2$). The lysates were centrifuged at 5000× *g* for 30 min to remove unbroken cells. The remaining supernatants were centrifuged at 45,000× *g* for 1 hr to pellet membranes. Crude membranes were suspended in 20 mL 0.5 % Sarkosyl in 20 mM sodium phosphate and shaken horizontally at 200 rpm for 30 min at room temperature. The crude membrane samples were centrifuged at 45,000× *g* for 1 hr to pellet the OM. The OM pellets were washed in ice-cold sodium phosphate and recentrifuged.

To prepare for mass spectrometry, samples were treated sequentially with urea, TCEP, iodoactin-amide, lysl endopeptidase, trypsin, and formic acid. Peptides were then desalted by HPLC with a Microm Bioresources C8 peptide macrotrap (3 mm × 8 mm). The digested samples were subjected to LC-MS/MS analysis on a nanoflow LC system, EASY-nLC 1200 (Thermo Fisher Scientific) coupled to a QExactive HF Orbitrap mass spectrometer (Thermo Fisher Scientific, Bremen, Germany) equipped with a Nanospray Flex ion source. Samples were directly loaded onto a PicoFrit column (New Objective, Woburn, MA) packed in-house with ReproSil-Pur C18AQ 1.9 µm resin (120 Å pore size, Dr Maisch, Ammerbuch, Germany). The 20 cm × 50 µm ID column was heated to 60 °C. The peptides were separated with a 120 min gradient at a flow rate of 220 nL/min. The gradient was as follows: 2–6% solvent B (7.5 min), 6–25% solvent B (82.5 min), and 25–40% solvent B (30 min) and to 100 % solvent B (9 min). Solvent A consisted of 97.8 % $H_2O$, 2 % acetonitrile, and 0.2 % formic acid and solvent B consisted of 19.8 % $H_2O$, 80 % ACN, and 0.2 % formic acid. The QExactive HF Orbitrap was operated in data-dependent mode with the Tune (version 2.7 SP1build 2659) instrument control software. Spray voltage was set to 2.5 kV, S-lens RF level at 50, and heated capillary at 275 °C. Full scan resolution was set to 60,000 at m/z 200. Full scan target was $3 \times 106$ with a maximum injection time of 15 ms. Mass range was set to 300–1650 m/z. For data-dependent MS2 scans, the loop count was 12, target value was set at $1 \times 105$, and intensity threshold was kept at $1 \times 105$. Isolation width was set at 1.2 m/z and a fixed first mass of 100 was used. Normalized collision energy was set at 28. Peptide match was set to off, and isotope exclusion was on. Data acquisition was controlled by Xcalibur (4.0.27.13) and all data was acquired in profile mode. The mass spectrometry proteomics data have been deposited to the ProteomeXchange Consortium via the PRIDE (*Perez-Riverol et al., 2019*) partner repository with the dataset identifier PXD020577.

The raw data was analyzed using MaxQuant (version 1.6.1.0). Spectra were searched against the *S. oneidensis* sequences from UniProt as well as a contaminant protein database. Trypsin was specified as the digestion enzyme and up to two missed cleavages were allowed. Carbamidomethylation of cysteine was specified as a fixed modification and protein N-terminal acetylation as well as methionine oxidation were specified as variable modifications. Precursor mass tolerance was 4.5 ppm after recalibration within MaxQuant. Spectrum, peptide, and protein scores were thresholded to achieve a 1 % false discovery rate at each level. False discovery rates were estimated using a target-decoy approach. Label-free quantitation and match-between-runs was enabled. Missing values were imputed from a normal distribution centered near the limit of quantitation. Log fold change and p-values were computed from three biological replicates in each condition.

## Bioinformatics

Putative BAR domain SO_1507 (BdpA) was identified in search of annotation terms of *S. oneidensis* MR-1. The conserved domain database (CDD-search) (NCBI) was accessed to identify the PSSM of the specific region of SO_1507 that represented the BAR domain (amino acid residues at positions 276–421). The domain prediction matched to BAR superfamily cl12013 and specifically to the family member BAR cd07307. PSI-BLAST (*Altschul et al., 1997*) was used to identify homologs of BdpA that were not annotated as containing a BAR domain. The *S. oneidensis* MR-1 protein AAN54568 sequence from positions 276 to 451 was used as the initial query. The 24 subject proteins with expectation values below the threshold of 0.005 were used to construct the matrix for a second iteration which returned an additional 28 proteins. Further iterations did not identify any more homologs. LOGICOIL multi-state coiled-coil oligomeric state prediction was used to predict the presence of coiled-coils within BdpA (*Vincent et al., 2013*). SignalP 6.1 was used to detect the presence of the signal peptide and cellular localization of BdpA (*Nielsen, 2017*). The ab initio predicted structure of BdpA was generated by trRosetta (*Yang et al., 2020*) and visualized in the PyMOL Molecular Graphics System, version 2.0 (Schrödinger, LLC). The BdpA dimer structure model was predicted using the RosettaDock and docking2 software on ROSIE (*Lyskov and Gray, 2008*; *Chaudhury et al., 2011*; *Lyskov et al., 2013*).

Alignments and phylogenies of the 52 BdpA homologs with the 23 BAR superfamily HMM seed sequences, as well as the alignments and phylogenies for BdpA homologs with representative BAR domain subtypes were constructed in MEGA 7. MUSCLE was used to align these protein sequences, and maximum likelihood phylogenies were inferred using the Le-Gascuel (LG + G) substitution matrix (*Le and Gascuel, 2008*)}. Initial trees for the heuristic search were obtained automatically by applying

Neighbor-Join and BioNJ algorithms to a matrix of pairwise distances estimated using a JTT model, and then selecting the topology with superior log likelihood value. A discrete gamma distribution was used to model evolutionary rate differences among sites as indicated in figure legends. All positions with less than 85 % site coverage were eliminated.

Protein sequences of the five *Shewanella* BdpA homologs, representative bacterial proteins containing putative amphipathic helices (FtsA, MamY, and PmtA), and the 61 most diverse representative sequences representing the conserved domain in the CDD were trimmed to only the regions aligning to the BAR domain (or to regions previously identified as relevant for membrane association in FtsA, MamY, and PmtA) and aligned using the COBALT webserver, with parameters adjusted for more distantly related sequences by using word size = 3 (*Supplementary file 2*). A phylogenetic tree was constructed with the IQ-TREE webserver (*Trifinopoulos et al., 2016*) using the Le-Gascuel+ frequencies model with four discreet gamma categories. Branching confidence values were obtained with 1000 fast approximate bootstraps. The generalized midpoint optimization strategy was used to select a root.

## Ferrihydrite reduction

In order to test the iron reduction of the different strains, overnight cultures of each strain were pelleted and re-suspended in SDM to an OD of 0.1 and incubated in a 96-well plate inside an anaerobic chamber (Coy Laboratories, Grass Lake, MI) with 25 mM ferrihydrite. Samples for each time point were diluted 1:10 in 0.5 N hydrochloric acid. The acid fixed samples were then diluted 1:10 in ferrozine reagent (Thermo Fisher Scientific, 2 g/L in 500 mM HEPES buffer pH 7) and read immediately at an absorbance of 562 nm. Standards were prepared using iron sulfate dissolved in 0.5 N hydrochloric acid.

## Dynamic light scattering

Distributions of vesicle diameters were measured with Wyatt Technology's Möbiu $\zeta$ DLS instrument with DYNAMICS software for data collection and analysis. Data was collected using a 0–50 mW laser at 830 nm. The scattered photons were detected at 90°. Measurements were recorded from 11 biological replicates (independent cultures) for WT OMVs, 9 replicates for $\Delta bdpA$ OMVs, and 3 replicates for $\Delta bdpA+ bdpA$ OMVs prepared as described above. Mobius software analyzed the population of particles to generate a table with binned diameters and the percentage of particles at each diameter. Histograms were generated by plotting the average percentage of particles (intensity) for the whole population at each binned diameter.

In order to compare the average vesicle size from each sample, a weighted average was computed so that diameter bins that had the greatest number of vesicles would be accurately represented in the final average weight of the population. The product of each diameter was multiplied by its percentage in the population; these products were added together for each sample, divided by the sum of weights. The weighted diameters per replicate were then averaged for each genotype. Statistical significance was determined by Student's t-test, and error bars represent standard deviation. The F-test was used to compare the variance in the distribution of OMVs between strains. When an unequal variance was detected, we applied the Welch's post-correction to further compare means.

## Perfusion flow microscopy

For perfusion flow experiments, *S. oneidensis* WT and $\Delta bdpA$ strains were pre-grown aerobically from frozen (–80 °C) stock in 10 mL of LB broth in a 125 mL flask overnight at 30 °C and 225 rpm. The next day, the stationary phase (OD$_{600}$ 3.0–3.3) preculture was used to inoculate 1:100 into 10 mL of fresh LB medium in a 125 mL flask. After ~6 hr at 30 °C and 225 rpm, when the OD$_{600}$ was 2.4 (late log phase), 5 mL of cells were collected by centrifugation at 4226× $g$ for 5 min and washed twice in SDM wash media (SDM without added MgCl$_2$, CaCl$_2$, and Fe-NTA). Measurements were recorded from three biological replicates (independent cultures) for each strain on separate days.

The perfusion chamber, microscope, and flow medium described previously (*Chong et al., 2019*; *Subramanian et al., 2018*; *Pirbadian et al., 2014*) were used here. For each imaging experiment, the perfusion chamber was first filled with SDM flow medium, then <1 mL of washed cells were slowly injected for a surface density of ~100–300 cells per 112 × 112 μm field of view on a Nikon Eclipse Ti-E inverted microscope with the NIS-Elements AR software. Cells were allowed to attach for 5–15 min

on the coverslip before perfusion flow was resumed at a volumetric flow rate of 6.25 ± 0.1 µL/s. Cells were visualized with the red membrane stain FM 4–64 FX which was added to the flow medium (0.25 µg/mL of flow medium). At least five random fields of view for each biological replicate were repeatedly imaged at 5 min intervals over the course of 5 hr, creating a time-lapse series similar to time-lapse microscopy previously performed (*Chong et al., 2019*; *Subramanian et al., 2018*; *Pirbadian et al., 2014*). A total of 2607 WT and 2943 Δ*bdpA* cells (from three independent biological replicate experiments per strain) were manually counted and inspected for OMVs and OMEs, as done previously for WT (*Chong et al., 2019*). For each independent experiment, the proportion of cells producing either type of membrane feature over the course of the experiment was calculated by dividing the number of cells for which a membrane feature was observed by the number of total cells for each strain; the proportions from each independent experiment were averaged for each strain to obtain the mean proportions for each strain that were plotted in bar graphs. A random subset of time-lapse microscopy videos from each replicate (1273 WT cells and 1317 Δ*bdpA* cells) were reanalyzed to specifically identify cells that made large vesicles, whether individual vesicles (directly associated with the stained membrane of the cell body) or noted as part of a vesicle chain. Considering diffraction-limited resolution in widefield fluorescence microscopy, large vesicles were defined as those where an unstained interior could clearly be resolved and were typically found to be >300 nm. Estimates of vesicle diameter were determined using the line measurement tool in ImageJ. For each independent experiment, the proportion of cells producing large vesicles was calculated by dividing the number of cells producing large vesicles by the total number of cells; the proportions from each independent experiment were averaged for each strain to obtain the mean proportions for each strain that were plotted in bar graphs. Statistical significance between strains was determined using the Pearson's chi-squared test for comparing proportions.

## Imaging of static cultures

*S. oneidensis* WT, Δ*bdpA*, and Δ*bdpA+ bdpA* strains were grown in LB medium (supplemented with 50 µg/mL kanamycin for Δ*bdpA+ bdpA*) overnight, washed twice with SDM, and diluted to an $OD_{600}$ of 0.05 in 1 mL of SDM (supplemented with 50 µg/mL kanamycin and 12.5 µM DAPG for Δ*bdpA+ bdpA*). Three independent biological replicates (cultures) were included for each strain. To visualize the cell membranes, 100 µL of diluted culture was labeled with 1 µL 1 M FM 4–64. After staining, 10 µL of the labeled cell suspension was gently pipetted onto Lab-Tek chambered #1 cover glass (Thermo Fischer Scientific). On average, uninduced membrane extension formation could be observed 45 min after deposition of the cell suspension onto the chambered cover glass surface. After 3 hr, images were collected at 0.27–0.63 s intervals over 20 s per field of view to capture the dynamics of the rapidly moving OMEs and ensure accurate counting of OMEs as they traverse focal planes. These single frame time series images were collected of either a 50.71 µm by 50.71 µm (2 × zoom) or a 20.28 µm by 20.28 µm (5 × zoom) field of view on a Zeiss LSM 800 confocal microscope with a Plan-Apochromat 63 ×/1.4 numerical aperture oil immersion M27 objective. Widefield fluorescence images were taken using an LED-Module 511 nm light source at 74.2 % intensity with 583–600 nm filters and a 91 He CFP/YFP/mCherry reflector. Excitation and emission peaks were 506 and 751 nm, respectively. Images were collected using a Hamamatsu camera with a 250 ms exposure time. Images were recorded using the Zeiss Zen software (Carl Zeiss Microscopy, LLC, Thornwood, NY). All cells were counted manually and categorized as either with extension or without extension (a total of 2444 cells from WT, 4378 cells from Δ*bdpA*, and 3354 cells from Δ*bdpA+ bdpA*). The proportion of cells with OMEs was calculated by dividing the number of cells with extensions by the total number of cells per strain. Statistical significance between the mean proportion of each strain was determined by two-tailed Student's t-test.

To test for both planktonic production of OMEs in WT *S. oneidensis* (WT + *bdpA*) and expression of BdpA in *M. atlanticus* (CP1+ *bdpA*) and *E. coli* (BL21+ *bdpA*), overnight cultures for each strain were diluted in either SDM for *S. oneidensis*, BB for *M. atlanticus*, or LB for *E. coli* to an $OD_{600}$ of 0.05 and induced with the indicated concentration of DAPG for 1 hr at 30 °C (or 37 °C for *E. coli*) with 200 rpm shaking agitation. Prior to pipetting, ~ 1 cm of the p200 pipette tip was trimmed to minimize shear forces during transfer. A 100 µL aliquot of each culture was labeled with 1 µL 1 M FM 4–64 as before, 10 µL deposited onto 22 mm × 22 mm No. 1 cover glass (VWR), and sealed onto glass slides with clear acrylic nail polish to restrict mobility in the Z direction and facilitate immediate imaging of

OMEs produced during planktonic induction of BdpA. Each sample was imaged immediately after mounting onto the glass slides. Imaging experiments were performed on at least three individual biological replicate experiments per strain, and are representative images from 5 to 10 fields of view per replicate from 700 cells for *S. oneidensis* WT, 472 cells for *S. oneidensis* WT + *bdpA*, 4041 cells for *M. atlanticus* CP1 WT, 150 cells for *M. atlanticus* CP1+ *bdpA*, 2190 cells for *E. coli* BL21 WT, and 2623 cells for *E. coli* BL21+ *bdpA*. As before, the proportion of cells producing either type of membrane feature was calculated by dividing the number of cells for which a membrane feature was observed by the number of total cells for each strain; the proportions from each independent experiment were averaged for each strain to obtain the mean proportions for each strain that were plotted in bar graphs. The proportion of cells associating with OMEs observed from each biological replicate culture was recorded, as well as if the associated OME resembled either a tubule-like or web-like structure. Tubule-like OMEs were defined as narrow, unbranching membrane extensions. Cells that were associated with a branching, reticular membrane were counted as producing a web-like OME. Statistical significance of the proportions of cells associated to each of the OME phenotypes between strains was determined by Welch's t-test. Confocal images were taken by a Zeiss LSM 800 confocal microscope with a Plan-Apochromat 63 ×/1.4 numerical aperture oil immersion M27 objective. FM 4–64 fluorescence was excited at 506 nm: 0.20 % laser power. Fluorescence emission was detected from 592 to 700 nm using the LSM 800 GaAsP-Pmt2 detector. High-resolution confocal fluorescence images of CP1+ *bdpA* and BL21+ *bdpA* OMEs were collected using the Zeiss LSM 800 Airyscan detector module for the confocal microscope with a Plan-Apochromat 63 /1.4 numerical aperture oil immersion M27 objective, 10 × post-objective magnification, and Airyscan image processing. Images are Airyscan-processed maximum intensity projections of a z-stack image series over a 1.32–2.64 μm z-stack height at 0.18–20 μm intervals. Time course Airyscan images of BL21+ *bdpA* OME elongation were recorded of a single cell every 30 min for an hour. *S. oneidensis* OMEs could not be recorded by Airyscan due to rapid movement.

## Cryogenic transmission electron microscopy

*Shewanella* strains were streaked onto LB plates with or without kanamycin and allowed to incubate 3 days on a benchtop. The night before freezing, individual colonies were inoculated into 3 mL LB±kanamycin and incubated at 30 °C overnight with 200 rpm shaking. The following morning optical densities of the cultures were measured at 600 nm and adjusted to a final $OD_{600}$ of 1. Cells were pelleted at 8000 rpm for 3 min for buffer exchange/washes. For the Δ*bdpA* + *bdpA* cells, 12.5 μM DAPG was added. A freshly glow discharged 200 mesh copper grid with R2/1 Quantifoil carbon film was placed into a concavity slide. Approximately 150 μL of a 1:10 dilution of the cell suspensions, with or without the inducer, was added to cover the grid. A glass coverslip was then lowered onto the concavity to exclude air bubbles. The edges of the coverslip were then sealed with nail polish to prevent media evaporation. The slide assembly was then incubated in a 30 °C incubator for 90 min or 3 hr. Immediately prior to plunge freezing, the top coverslip was removed by scoring the nail polish with a razor blade. TEM grids with cells were gently retrieved with forceps and loaded into a Leica grid plunge for automated blotting and plunging into $LN_2$-cooled liquid ethane. Vitrified grids were transferred to an $LN_2$ storage dewar. Imaging of frozen samples was performed on either a Titan (Thermo Fisher Scientific) microscope equipped with a Gatan Ultrascan camera and operating at 300 kV or a Talos (Thermo Fisher Scientific) equipped with a Ceta camera and operating at 200 kV. Images were acquired at 10,000× to 20,000× magnification and were adjusted by bandpass filtering. Unfixed OMEs were sorted based on appearance into four categories as described in the main text, that is, tubules, narrow chains, irregular chains, or blebs/bulges. At the 90 min time point, phenotypes were documented from observations of 14 WT, 12 Δ*bdpA*, and 41 Δ*bdpA* + *bdpA* OMEs over three separate biological replicates, with two technical replicates of each strain per biological replicate. At the 3 hr time point, phenotypes were documented from observations of 31 WT, 13 Δ*bdpA*, and 3 Δ*bdpA* + *bdpA* OMEs over three separate biological replicates, with two technical replicates of each strain per biological replicate. For the 2 hr planktonic induction of BdpA in the WT + *bdpA* strain, nine total OMEs were observed across two separate biological replicates. The proportion of membrane phenotypes was calculated by summing the number of instances a phenotype was observed across all replicates for a given strain and dividing by the total number of OMEs observed for that strain. Representative images only are given for the 90 min time point and for the WT + p452 bdpA strain.

## Acknowledgements

We thank Dr Jeffrey Gralnick for helpful discussions and advice, as well as the *S. oneidensis* JG1194 strain; Dr Adam Meyer and Dr Chris Voigt for the DAPG-inducible Marionette promoter; Dr Annie Moradian and Dr Mike Sweredoski and the California Institute of Technology Proteome Exploration Lab for useful discussions on the preparation and analysis of proteomics data. Some of the cryo-TEM work was done in the Beckman Institute Resource Center for Transmission Electron Microscopy at Caltech. This work was supported by the United States Department of Defense Synthetic Biology for Military Environments (SBME) Applied Research for the Advancement of Science and Technology Priorities (ARAP) program. DP and AEM were partially supported by the U.S. Army via the Surface Science Initiative Program (PE 0601102 A Project VR9) at the Combat Capabilities Development Command (CCDC) Chemical Biological Center. Work in ME-N's lab was supported by the U.S. Office of Naval Research Multidisciplinary University Research Initiative Grant No. N00014-18-1-2632. LAZ was partially supported by the National Science Foundation grant DEB-1542527. GWC was also supported by the National Science Foundation Graduate Research Fellowship Program (grant DGE1418060). SX was supported by the Division of Chemical Sciences, Geosciences, and Biosciences, Office of Basic Energy Sciences of the U.S. Department of Energy through grant DE-FG02-13ER16415. Work in GJJ's lab was supported by the National Institute of Health (GM122588 to GJJ).

## Additional information

### Competing interests

Daniel A Phillips: along with SG holds the patent US10793865B2 on "Transferrable mechanism of generating inducible, BAR domain protein-mediated bacterial outer membrane extensions". Sarah M Glaven: along with DP holds the patent US10793865B2 on "Transferrable mechanism of generating inducible, BAR domain protein-mediated bacterial outer membrane extensions". The other authors declare that no competing interests exist.

### Funding

| Funder | Grant reference number | Author |
| --- | --- | --- |
| U.S. Department of Defense | | Sarah M Glaven |
| Office of Naval Research | N00014-18-1-2632 | Mohamed Y El-Naggar |
| National Science Foundation | DEB-1542527 | Mohamed Y El-Naggar |
| U.S. Department of Energy | DE-FG02-13ER16415 | Mohamed Y El-Naggar |
| National Institute of General Medical Sciences | GM122588 | Grant J Jensen |
| U.S. Army Combat Capabilities Development Command | PE 0601102A Project VR9 | Aleksandr E Miklos |

The funders had no role in study design, data collection and interpretation, or the decision to submit the work for publication.

### Author contributions

Daniel A Phillips, Conceptualization, DP and LZ conceived the study independently then combined projects when complementary data on BdpA was discovered., Data curation, Formal analysis, Funding acquisition, Investigation, Methodology, Project administration, Resources, Supervision, Validation, Visualization, Writing – original draft, Writing – review and editing; Lori A Zacharoff, Conceptualization, Data curation, Formal analysis, Funding acquisition, Investigation, Methodology, Project administration, Resources, Supervision, Validation, Visualization, Writing – original draft, Writing – review and editing; Cheri M Hampton, Formal analysis, Investigation, Methodology, Visualization, Writing – review and editing; Grace W Chong, Data curation, Formal analysis, Investigation, Methodology,

Visualization, Writing – review and editing; Anthony P Malanoski, Data curation, Formal analysis, Investigation, Methodology, Visualization, Writing – original draft, Writing – review and editing; Lauren Ann Metskas, Investigation, Methodology, Visualization, Writing – review and editing; Shuai Xu, Lina J Bird, Resources, Writing – review and editing; Brian J Eddie, Data curation, Formal analysis, Investigation, Validation, Writing – original draft, Writing – review and editing; Aleksandr E Miklos, Funding acquisition, Investigation, Software, Visualization; Grant J Jensen, Funding acquisition, Supervision, Writing – review and editing; Lawrence F Drummy, Investigation, Methodology, Supervision, Validation, Visualization, Writing – review and editing; Mohamed Y El-Naggar, Sarah M Glaven, Formal analysis, Funding acquisition, Project administration, Resources, Supervision, Writing – original draft, Writing – review and editing

## Author ORCIDs
Daniel A Phillips http://orcid.org/0000-0003-2759-5246
Lori A Zacharoff http://orcid.org/0000-0001-8657-0968
Cheri M Hampton http://orcid.org/0000-0003-0069-8712
Grace W Chong http://orcid.org/0000-0003-1369-1405
Anthony P Malanoski http://orcid.org/0000-0001-6192-888X
Lauren Ann Metskas http://orcid.org/0000-0002-8073-6960
Shuai Xu http://orcid.org/0000-0001-8849-7506
Lina J Bird http://orcid.org/0000-0003-4127-4756
Brian J Eddie http://orcid.org/0000-0002-3559-3892
Aleksandr E Miklos http://orcid.org/0000-0001-6375-2304
Grant J Jensen http://orcid.org/0000-0003-1556-4864
Lawrence F Drummy http://orcid.org/0000-0002-6452-5768
Mohamed Y El-Naggar http://orcid.org/0000-0001-5599-6309
Sarah M Glaven http://orcid.org/0000-0003-0857-3391

## Decision letter and Author response
Decision letter https://doi.org/10.7554/eLife.60049.sa1
Author response https://doi.org/10.7554/eLife.60049.sa2

## Additional files

### Supplementary files
• Supplementary file 1. Protein enrichment within the outer membrane vesicle (OMV) proteome relative to the proteome of the *Shewanella oneidensis* outer membrane (OM).

• Supplementary file 2. Alignment of BdpA homologs with bacterial membrane curvature associated proteins and eukaryotic Bin/Amphiphysin/RVS (BAR) domains.

• Transparent reporting form

### Data availability
The mass spectrometry proteomics data have been deposited to the ProteomeXchange Consortium via the PRIDE [1] partner repository with the dataset identifier PXD020577.

The following dataset was generated:

| Author(s) | Year | Dataset title | Dataset URL | Database and Identifier |
|---|---|---|---|---|
| Zacharoff LA, El-Naggar MY | 2021 | Shewanella oneidensis outer membrane and outer membrane vesicles | https://www.ebi.ac.uk/pride/archive/projects/PXD020577 | PRIDE, PXD020577 |

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
