## [Decision Letter]

**Acceptance summary:**

In this manuscript the authors propose the identification of a novel protein involved in outer membrane remodeling, named BdpA (BAR domain-like protein A). They propose that BdpA has a conserved role in membrane curvature control during formation of outer membrane vesicle (OMV) and of outer membrane extension (OMEs) in *Shewanella oneidensis*. The authors provide evidence that heterologous expression of BdpA promotes formation of OMEs in other bacteria (namely in *E. coli*), and that, BdpA is sufficient to induce OME-like structures when expressed in in conditions where OMEs are normally not formed. In eukaryotes proteins containing BAR domains are important for shaping membrane curvature. BdpA has a low homology to eukaryotic BAR-domain proteins and given that like the eukaryotic proteins, BdpA has a role in altering membrane architecture in bacteria, the authors suggest that BdpA is Bacterial Membrane Sculpting Protein with BAR Domain-like Activity.

**Decision letter after peer review:**

Thank you for submitting your article "A Prokaryotic Membrane Sculpting BAR Domain Protein" for consideration by *eLife*. Your article has been reviewed by 3 peer reviewers, one of whom is a member of our Board of Reviewing Editors, and the evaluation has been overseen by Gisela Storz as the Senior Editor. The following individuals involved in review of your submission have agreed to reveal their identity: Arash Komeili (Reviewer #2); Gemma Reguera (Reviewer #3).

The reviewers have discussed the reviews with one another and the Reviewing Editor has drafted this decision to help you prepare a revised submission.

As the editors have judged that your manuscript is of interest, but as described below that additional experiments are required before it is published, we would like to draw your attention to changes in our revision policy that we have made in response to COVID-19 (https://elifesciences.org/articles/57162). First, because many researchers have temporarily lost access to the labs, we will give authors as much time as they need to submit revised manuscripts. We are also offering, if you choose, to post the manuscript to bioRxiv (if it is not already there) along with this decision letter and a formal designation that the manuscript is "in revision at *eLife*". Please let us know if you would like to pursue this option.

Summary:

In this manuscript the authors propose the identification of a novel protein involved in outer membrane remodelling, named BdpA (BAR domain-like protein A). According to the proposed model BdpA has a conserved role in membrane curvature control during formation of outer membrane vesicle (OMV) and of outer membrane extension (OMEs) in Shewanella oneidensis. The authors also provide evidence that heterologous expression of BdpA promotes formation of OMEs in other bacteria (namely in *E. coli*), and that, BdpA is sufficient to induce OME-like structures when expressed in in conditions where OMEs are normally not formed. In eukaryotes proteins containing BAR domains are important for shaping membrane curvature. Given the homology of BdpA to eukaryotic BAR-domain proteins, the authors suggest that BdpA and its homologs define the first prokaryotic family of BAR proteins or pBARs, with eukaryotic-like roles in membrane curvature modulation.

Overall, the reviewers think that this is a very interesting study, and provided that further support is obtained to substantiate the proposed model the reviewers agree that the findings described here tackle a number of significant questions of broad interest. However, the reviewers also think that the evidence provided in this manuscript still does not fully support the conclusion that BdpA protein is involved in membrane curvature control as the eukaryotic proteins containing the BAR domains.

We have complied a list of comments that we hope will help the authors address the concerns of the reviewers to obtain stronger support for the function of BdpA.

Essential revisions:

1. The reviewers are concerned that some of the conclusions are based on qualitative observations of microscopy analysis of OMVs and OMEs, and quantitative analyses are lacking to validate qualitative observations. The reviewers propose that the data should be re-analyzed to obtain quantitative results. Specifically, a size distribution analyses could be applied to some microscopy data. Also note that the microcopy methods are poorly described, and as the calculation methods used not fully available it is difficult to understand if the appropriated methods were used. Please specify how many cells were examined microscopically and how many biological replicates (cultures) were used in each experiment.

2 Statistical analyses were not always the most accurate. In figure 2 unpaired t-test was used for samples that have high variance, this approach may inflate the statistical difference between the strains. For figure 2 a histogram of size distribution analyses could be shown for each strain.

3. The reviewers are concerned that the proteomic data is not clear enough to conclude that the BdpA protein is localized to or enriched in OMV/OME. Could the results be complemented with some other method to confirm BdpA localization? The reviewers are particularly concerned by the fact that a large number of proteins were identified in the OMV fraction. Could it be that some the OMV/OME fractions were contaminated? What controls were used to ensure that the purification procedure was working effectively? Could the data be strengthened by some quality control analyses to determine how many of those proteins are actually predicted to localize to the outer membrane and periplasm? From the methods it seems that the culture conditions used to prepare the OM versus OMV were different, is this so? If yes, why were the culture conditions different? This could affect protein expression? Please include the detailed growth conditions in the method section.

4. The conclusion that BdpA is a BAR-domain protein is largely based on homology. The supplementary information file includes homology models that show striking similarity with eukaryotic BAR proteins. However, as the authors state, BdpA barely meets the cutoff for a BAR-domain protein. The results with the phenotype of the BdpA mutant, complementations and sufficiency data provide good support to the functional role of BdpA in membrane remodelling. However, the effect of BdpA on membrane stability could be indirect or the result of binding to outer membrane features in a manner distinct from other BAR proteins. Could these results be strengthened with some biochemical corroboration of its activity on membranes or structural data to confirm its relationship to eukaryotic Bar domain proteins? Or structural data to confirm its relationship to eukaryotic BAR domain proteins?

5. The reviewers propose that the paper would be strengthened with the addition of topological studies in OMVs and OMEs. The reviewers had problems in reconciling the presence of a galactose-binding domain in BdpA and LPS sugar binding. The authors hypothesize that the putative Galactose-binding domain of BdpA mediates binding to LPS. However, it is also possible that it binds to peptidoglycan components. This would suggest that the proteins interact with the periplasmic side of the outer membrane rather than coat the OMV to promote OMV formation and release (which one could assume based on the role of some eukaryotic BARs). The addition of topological studies (or some biochemical approach) could make these models less speculative strengthening the conclusions.

6. Heterologous expression of BdpA in other bacteria provides important compelling arguments for its central role in producing OMEs. However, the imaging data provided do not provide the clearest evidence for induction of OMEs in M. atlanticus and *E. coli*. This is especially the case with the *E. coli* images. The extended web of staining in 4c does not resemble the tubules seen in S. oneidensis. It would be great to have some electron microscopy data and/or higher resolution fluorescence images of these bacteria as corroborating evidence. Additionally, only a few cells are shown and quantification of the proportion of cells with OMEs is needs. Thus, as already discussed in point 1, quantitative analyses could improve this important point.

[Editors' note: further revisions were suggested prior to acceptance, as described below.]

Thank you for submitting the revised version of your article "A Prokaryotic Membrane Sculpting BAR Domain Protein" for consideration by *eLife*. The revised version of your article, and your answers to the reviewers' comments have been reviewed by 2 peer reviewers, and Karina Xavier as Reviewing Editor, the evaluation has been overseen by Gisela Storz as the Senior Editor. The following individuals involved in review of your submission have agreed to reveal their identity: Arash Komeili (Reviewer #2); Gemma Reguera (Reviewer #3).

The reviewers and the Reviewing Editor have discussed their reviews with one another, and the Reviewing Editor has drafted this to help you prepare a revised submission.

Essential revisions:

Overall, the reviewers and the Reviewing Editor have agreed that the authors have done a very good job in replying to their previous concerns and that the changes made by the authors have increased the quality of the paper very significantly. Nonetheless, based on the individual comments of the reviewers below and the discussion among them through the consultation section the reviewers thought that there are still some essential points that need to be modified before accepting the paper in *eLife*.

1) An important point is related to how to classify the BdpA sculpting protein studied here. The reviewers think that this protein should be called a Bacterial Membrane Sculpting Protein as opposed to a "prokorayotic" because there is no solid support for the presence of such a protein in Archaea. Additionally, upon discussion there are still concerns that the homology data of BdpA with BAR domain proteins is not very strong, especially taking into account that mistakes have been previously made with other proteins (eg. MamY).

We would like to propose that the authors put emphasis on the fact that the paper shows that BdpA has an important role in Outer membrane Vesicle formation in bacteria. Additionally, as the paper shows strong experimental support that BdpA is working as a Bacterial Membrane Sculpting Protein with a BAR-like mechanism, we propose that more emphasis is put on that than on the homology. Therefore, we propose that the title is change to something like A Bacterial Membrane Sculpting Protein with a BAR-like mechanism, or simply A Bacterial Membrane Sculpting Protein.

The text should also be changed to give more emphasis that the classification of BdpA as sculpting protein with a BAR-like mechanism is based on the observed phenotypes, and on structural similarities to eukaryotic BAR proteins, more than only homology.

2) As it is possible that the BdpA effects observed are indirect (for example through indirect interactions with the peptidoglycan or LPS sugar components), it is important that the authors acknowledge in the paper that future required to address this possibility. Additionally, as FtsA is related to BdpA and eukaryotic BARs (FIGURE 5), we recommend that the authors discuss the possibility that FtsA and BdpA have redundant functions in cell division, namely by stabilizing FtsZ (see reviewer #3 comments).

3) Please revised the statistical analysis of Figure 3D. Reviewer #3 is suggesting that it might be better to use these results for depicting the variations in OME structure rather than definitive assessment of their phenotypes in various strains. (see specific comment 1 of Reviewer #3)

*Reviewer #2:*

The authors have provided extensive data and methodology that answer many of my original questions. I still have a few questions and concerns:

1. Some of the statistics are not convincing. If I am reading the legend correctly, Figure 3D numbers are based on a very small number of samples "Frequency of OME phenotypes observed with Cryo-TEM relative to the total number of OMEs observed from each strain. Phenotypes were documented from observations of 14 WT (p = 0.009), 12 ΔbdpA (p = 0.0001), and 41 ΔbdpA + bdpA (p = 2.8x10^-6^) OMEs at the 90 minute time point, and 31 WT, 13 ΔbdpA (p = 6.3x10^-9^), and 3 ΔbdpA + bdpA (p > 0.05)". At the extreme end, the 3 hour complemented bar graph is depicting 3 OMEs measured. I fully understand that it is difficult to obtain high number of images with cryoEM. Perhaps, this section is more useful for depicting the variations in OME structure rather than definitive assessment of their phenotypes in various strains. How much of the observed statistical differences are influenced by the low sampling numbers?

2. I still do not agree that "prokaryotic" is more appropriate than "bacterial." One of the primary criteria used to classify BdpA as a BAR domain protein is based on bioinformatic classification. Are there any similar archaeal proteins? If not, given the number of available archaeal genomes, the eukaryotic/prokaryotic separation seems arbitrary.

3. My other major concern is whether or not this protein is a BAR-domain protein. That is the main thrust of the paper from the title through much of the text. The effects on membrane stability are interesting but could still be indirect through interactions with the peptidoglycan or LPS sugar components. The homology to eukaryotic BAR domains is borderline. To be clear, this is not a criticism of the authors. They have done a fantastic job given the difficulties of the system. However, I am not convinced that BdpA can be called a BAR domain protein and in several instances (the title, abstract, etc) this claim is centered.

*Reviewer #3:*

This article identifies in Shewanella oneidensis MR-1 the first prokaryotic BAR domain protein, BdpA, and links its expression to the production of redox-active membrane vesicles and micrometer scale outer membrane extensions (OMEs) in this bacterium. The authors used various complementary approaches to link BdpA expression to the control of OMV size and scaffolding into OMEs. Furthermore, they identified BdpA homologs in other bacteria and heterologously express the protein in BAR-deficient hosts to induce OME production. This research provides much-needed mechanistic understanding of bacterial outer membrane remodeling. I had liked this work very much when I reviewed a previous version of the manuscript. The revisions added clarity to the methods, improved data presentation in the figures, and made for an enjoyable read. I commend the authors for taking the reviewers comments into consideration when revising the manuscript. They did a phenomenal job. I predict this article will establish the intellectual foundation for research in prokaryotic BAR proteins and will be cited extensively.

I only have one comment to share in case the authors want to add it to the discussion and/or find it useful for follow-up work. FtsA is related to BdpA and eukaryotic BARs (FIGURE 5). Given its role in stabilizing the FtsZ ring prior to the assembly of the divisome, FtsA and BdpA could have redundant functions in cell division. This could help explain why the complemented BdpA strain had growth defects (BdpA substitutes for FtsA but may not be able to interact productively with FtsZ or other proteins of the divisome).

---

## [Author Response]

Essential revisions:1. The reviewers are concerned that some of the conclusions are based on qualitative observations of microscopy analysis of OMVs and OMEs, and quantitative analyses are lacking to validate qualitative observations. The reviewers propose that the data should be re-analyzed to obtain quantitative results. Specifically, a size distribution analyses could be applied to some microscopy data.

It is unclear which microscopy the reviewers are specifically referring to here. We interpret this comment to refer to the perfusion flow experiments in Figure 2, as well as fluorescence microscopy in Figures 3 and 4.

We have reorganized Figure 2 where Figure 2c is now Figure 2b. Figure 2b presents quantitative data on the proportion of live cells from each strain that display some membrane feature (OME and/or OMV) over the course of the experiment. This experiment was performed to complement DLS analysis of purified OMVs. The advantage of the perfusion flow experiment is that it allows us to visualize the proportion of live cells that display membrane features over time and to categorize these features. Cell membrane features are dynamic and must be manually categorized by monitoring any changes using time-lapse imaging. For this reason, we imaged and manually categorized a total of 2,607 WT and 2,943 Δ*bdpA* cells by visually inspecting and counting the dominant feature of each cell over 5 hours. As with the DLS size distribution analysis of OMVs, no difference in the total number of OMVs was observed between strains. We also quantified OMEs and again observed no difference in the total number between strains. We hypothesized that, as seen with DLS, it was the size of the OMVs, and possibly the shape of OMEs, that is different between strains. Unfortunately, fluorescence microscopy does not allow for the resolution required to visualize the size range of OMVs determined by DLS or differences in OME morphology, therefore, we re-analyzed a subset of our data to quantify the proportion of cells that produced membrane associated large vesicles that could be clearly distinguished within the limits of resolution of our measurements, those with an approximate diameter of 300 nm or more. This limitation still allowed us to confirm that even in live cells, the mutant strain produces a greater proportion of larger OMVs. Cryo-electron microscopy presented in Figure 3e and 3f validate that indeed the morphology of OMEs in the mutant strain was different than the WT strain. We have updated the text to clarify our rationale for quantifying the total proportion of membrane features and large OMVs vs. a size distribution. We have revised Figure 2b to show the proportion of both OMVs and OMEs per strain whereas they had previously been grouped together. We also added a representative time-lapse video of both strains to show examples of the features that were quantified (Videos 1 and 2), and a representative image of these features from the Δ*bdpA* strain in Figure 2—figure supplement 3.

Figure 3

This figure has been completely reorganized. The figure caption and methods section have been revised to include the number of biological replicates, number of cells that were visualized / categorized as having an OME (or not) for all strains. In order to clarify, we have moved previous Supplemental Figure 2 to the main figure to highlight the quantification. In addition, we have clarified the methods and figure caption for quantification of the proportion of various OME phenotypes at 90 minutes and 3 hours for all strains reported in Figure 3. We have also included details on the statistical comparison in the figure caption and main text used to conclude that cryo-TEM images show a difference in OME phenotypes between strains and times. We have also moved images of OMEs taken at 90 minutes for the same strains from Supplemental Figure 3 to the main Figure 3 and added quantification / statistical support for these as well.

Figure 4 is discussed below in response to Essential revision 6.

Also note that the microscopy methods are poorly described, and as the calculation methods used not fully available it is difficult to understand if the appropriated methods were used.

We agree with the reviewers that microscopy methods needed to be described in more detail. We have expanded our description of the perfusion flow experiment and fluorescence imaging of static cultures in the Methods section.

Please specify how many cells were examined microscopically and how many biological replicates (cultures) were used in each experiment.

We have made significant changes to the methods section, figure captions and main text where appropriate to indicate the number of biological replicates, total number of fields imaged, and total number of cells imaged. To be clear, we consider a biological replicate to be an independent culture.

2. Statistical analyses were not always the most accurate. In figure 2 unpaired t-test was used for samples that have high variance, this approach may inflate the statistical difference between the strains.

We agree with the reviewers that the variance in the size distribution of vesicles between the wild type and Δ*bdpA* strains is unequal. To further emphasize this, we performed a F-test of equality of variance. The F ratio between the WT and mutant strains to compare variances was a high 38.75, with a significant P value of <0.00001, indicating that the variance is not equal. Therefore, we applied the Welch’s t-test to compare unpaired samples with unequal variance. In this case, there was no statistically significant difference between the mean vesicle size of the wild-type strain compared to the Δ*bdpA* strain. We have indicated this in the text but prefer to keep the original figure.

For figure 2 a histogram of size distribution analyses could be shown for each strain.

We have revised Figure 2a to show individual, binned, size distribution plots for each strain. We assume this addresses the reviewer’s comment, since a histogram would have resulted in a dimensional reduction of the data. Indeed, the revised figure more clearly shows that the mutant strain has a greater variance in the vesicle size distribution, supporting our hypothesis that the *S. oneidensis* BdpA protein is involved in membrane architecture. We thank the reviewer for the suggestion, which improved the clarity of the presentation.

3. The reviewers are concerned that the proteomic data is not clear enough to conclude that the BdpA protein is localized to or enriched in OMV/OME. Could the results be complemented with some other method to confirm BdpA localization?

We agree with the reviewers that other methods to determine the localization of BdpA on the membrane of live cells would be ideal. Many attempts were made to perform immunofluorescence labeling or fluorescently tagged versions of BdpA. In the immufluorescence labeling experiments, OMEs were sheared off of cells during numerous washes necessary to mitigate backround fluorescence, even after fixation. The fragile nature of OMEs was documented in Subramanian et al., 2018. We cannot provide this data at this time, but further attempts to label BdpA will be investigated in future studies.

The reviewers are particularly concerned by the fact that a large number of proteins were identified in the OMV fraction. Could it be that some the OMV/OME fractions were contaminated?

Regarding the quantity of proteins detected in the OMVs, the review by Lee *et al.* (2016) detailed extensively how OMVs are frequently packed with many proteins, and the quantity detected in our preparations is within an expected range. We have added this review and added a sentence about the proteomics to line 99. And a further sentence to methods, line 459, modified to clarify: “This more inclusive purification method was chosen to prevent biasing the results in favor of vesicles of specific sizes and increase the likelihood of isolating more structures, which might be a concern with density gradient centrifugation. As a result, co-purification of extracellular DNA, flagella, and pili was expected. This protocol was adapted from Perez-Cruz et al.” Good digestion and sensitive, optimally calibrated equipment can improve detection of proteins that exist at a very low abundance.

What controls were used to ensure that the purification procedure was working effectively?

Three biological replicates were used to help detect contaminants. The methods section on line 490 was updated to reiterate this. Similarly, neither the DLS data nor the cryoelectron tomography of the vesicle preparations show either sizes or structures consistent with whole cell contamination.

Could the data be strengthened by some quality control analyses to determine how many of those proteins are actually predicted to localize to the outer membrane and periplasm?

To strengthen support that the purification procedures worked effectively, we have included the volcano plot colored based on p-sortb protein localization predictions (Figure 1—figure supplement 1). Through this analysis, more cytoplasmic proteins were detected in the OM fraction than in the vesicle samples. This plot was originally not included due to the large number of proteins with no known localization and discrepancies between psortb and other sorting programs. We also updated the methods to include the online repository location of the proteomics dataset for further clarity: "The mass spectrometry proteomics data have been deposited to the ProteomeXchange Consortium via the PRIDE [Perez-Riverol, 2019] partner repository with the dataset identifier PXD020577".

From the methods it seems that the culture conditions used to prepare the OM versus OMV were different, is this so? If yes, why were the culture conditions different? This could affect protein expression? Please include the detailed growth conditions in the method section.

We have now clarified the growth conditions in the vesicle / OM preparations. The growth conditions were the same with the exception of flask size due to considerably less biomass required for the outer membrane preparation, as whole cells are considerably larger than OMVs.

4. The conclusion that BdpA is a BAR-domain protein is largely based on homology. The supplementary information file includes homology models that show striking similarity with eukaryotic BAR proteins. However, as the authors state, BdpA barely meets the cutoff for a BAR-domain protein. The results with the phenotype of the BdpA mutant, complementations and sufficiency data provide good support to the functional role of BdpA in membrane remodelling. However, the effect of BdpA on membrane stability could be indirect or the result of binding to outer membrane features in a manner distinct from other BAR proteins. Could these results be strengthened with some biochemical corroboration of its activity on membranes or structural data to confirm its relationship to eukaryotic Bar domain proteins? Or structural data to confirm its relationship to eukaryotic BAR domain proteins?

We agree it would certainly solidify BdpA’s relationship to the eukaryotic BAR domain proteins. Rather than using a homology-based model as before, new, ab initio predicted structures were generated with the latest trRosetta deep learning model, and dimerization states were predicted in a similar manner by docking2. In this way, the BdpA monomers are not aligned to a eukaryotic BAR domain dimer, biasing the results based once again on homology, but are instead iteratively tested for the lowest energy state. We believe that since this ab initio model resulted in a characteristic BAR domain protein-like structure, the models strengthen support for our hypothesis. The new models were added to the main Figure 1 to better highlight this, and predicted domains were illustrated by color coordination with Figure 1d to help orient the reader.

5. The reviewers propose that the paper would be strengthened with the addition of topological studies in OMVs and OMEs. The reviewers had problems in reconciling the presence of a galactose-binding domain in BdpA and LPS sugar binding. The authors hypothesize that the putative Galactose-binding domain of BdpA mediates binding to LPS. However, it is also possible that it binds to peptidoglycan components. This would suggest that the proteins interact with the periplasmic side of the outer membrane rather than coat the OMV to promote OMV formation and release (which one could assume based on the role of some eukaryotic BARs). The addition of topological studies (or some biochemical approach) could make these models less speculative strengthening the conclusions.

We agree with the reviewers that topological studies to determine the location of BdpA on the membrane would be ideal but were outside of the scope of this study. We’ve revised the paper to be less speculative on the topology and make no claims as of yet as to the biophysical mechanisms mediating OME/V formation. We’ve also added text on line 286 stating that “Additional biochemical and biophysical assays beyond the scope of this initial manuscript are needed to further elucidate the effect of BdpA on membrane sculpting.”

6. Heterologous expression of BdpA in other bacteria provides important compelling arguments for its central role in producing OMEs. However, the imaging data provided do not provide the clearest evidence for induction of OMEs in M. atlanticus and E. coli. This is especially the case with the *E. coli* images. The extended web of staining in 4c does not resemble the tubules seen in S. oneidensis. It would be great to have some electron microscopy data and/or higher resolution fluorescence images of these bacteria as corroborating evidence. Additionally, only a few cells are shown and quantification of the proportion of cells with OMEs is needs. Thus, as already discussed in point 1, quantitative analyses could improve this important point.

We agree, and performed additional experiments aimed at broader quantification of these phenotypes. We quantified the proportion of cells of each strain displaying each phenotype and added to Figure 4. We have also clarified the text and updated the methods to reflect the additional quantification.

[Editors' note: further revisions were suggested prior to acceptance, as described below.]

Essential revisions:Overall, the reviewers and the Reviewing Editor have agreed that the authors have done a very good job in replying to their previous concerns and that the changes made by the authors have increased the quality of the paper very significantly. Nonetheless, based on the individual comments of the reviewers below and the discussion among them through the consultation section the reviewers thought that there are still some essential points that need to be modified before accepting the paper in eLife.1) An important point is related to how to classify the BdpA sculpting protein studied here. The reviewers think that this protein should be called a Bacterial Membrane Sculpting Protein as opposed to a "prokorayotic" because there is no solid support for the presence of such a protein in Archaea.

We agree and have revised all mention of “prokaryotic” to say “bacterial” for this manuscript.

Additionally, upon discussion there are still concerns that the homology data of BdpA with BAR domain proteins is not very strong, especially taking into account that mistakes have been previously made with other proteins (eg. MamY).We would like to propose that the authors put emphasis on the fact that the paper shows that BdpA has an important role in Outer membrane Vesicle formation in bacteria. Additionally, as the paper shows strong experimental support that BdpA is working as a Bacterial Membrane Sculpting Protein with a BAR-like mechanism, we propose that more emphasis is put on that than on the homology. Therefore, we propose that the title is change to something like A Bacterial Membrane Sculpting Protein with a BAR-like mechanism, or simply A Bacterial Membrane Sculpting Protein.

We also agree with this, and have revised the title to “A Bacterial Membrane Sculpting Protein with BAR Domain-like Activity”.

The text should also be changed to give more emphasis that the classification of BdpA as sculpting protein with a BAR-like mechanism is based on the observed phenotypes, and on structural similarities to eukaryotic BAR proteins, more than only homology.

We have altered the abstract and the main text to highlight the mechanistic similarities rather than the homology. Likewise, we have changed all mention of BdpA as a BAR domain protein to a membrane sculpting protein with BAR domain-like activity or a putative BAR domain-like protein.

2) As it is possible that the BdpA effects observed are indirect (for example through indirect interactions with the peptidoglycan or LPS sugar components), it is important that the authors acknowledge in the paper that future required to address this possibility.

On line 421 the following sentence was added into the conclusion section within the text, “While the presence of the predicted galactose-binding domain-like region at the N-terminal end of the protein suggests possible localization to the outer membrane, we cannot rule out that membrane sculpting phenotypes observed with BdpA are an indirect consequence of binding to LPS or through association with periplasmic components, such as peptidoglycan, at this time.”

Additionally, as FtsA is related to BdpA and eukaryotic BARs (FIGURE 5), we recommend that the authors discuss the possibility that FtsA and BdpA have redundant functions in cell division, namely by stabilizing FtsZ (see reviewer #3 comments).

The following sentences were added into the main text at line 383: “Induction of BdpA expression in the Δ*bdpA* + *bdpA* strain displayed attenuated growth (Figure 2—figure supplement 1). It is possible that BdpA could have redundant activity with FtsA, another alpha-helical protein known to stabilize FtsZ during contractile ring formation and cell division (47). Crowding of BdpA at the membrane could potentially inhibit the ability of FtsA to interact with FtsZ. Further investigation into the structure and function of BdpA is needed to determine its role in cell division. Likewise, structural and in vitro biophysical data are required before confirming that this mechanistically similar protein represents a new class of bacterial BAR domain-containing proteins.”

3) Please revised the statistical analysis of Figure 3D. Reviewer #3 is suggesting that it might be better to use these results for depicting the variations in OME structure rather than definitive assessment of their phenotypes in various strains. (see specific comment 1 of Reviewer #3)

Answered further below.

Reviewer #2:The authors have provided extensive data and methodology that answer many of my original questions. I still have a few questions and concerns:1. Some of the statistics are not convincing. If I am reading the legend correctly, Figure 3D numbers are based on a very small number of samples "Frequency of OME phenotypes observed with Cryo-TEM relative to the total number of OMEs observed from each strain. Phenotypes were documented from observations of 14 WT (p = 0.009), 12 ΔbdpA (p = 0.0001), and 41 ΔbdpA + bdpA (p = 2.8x10^-6^) OMEs at the 90 minute time point, and 31 WT, 13 ΔbdpA (p = 6.3x10^-9^), and 3 ΔbdpA + bdpA (p > 0.05)". At the extreme end, the 3 hour complemented bar graph is depicting 3 OMEs measured. I fully understand that it is difficult to obtain high number of images with cryoEM. Perhaps, this section is more useful for depicting the variations in OME structure rather than definitive assessment of their phenotypes in various strains. How much of the observed statistical differences are influenced by the low sampling numbers?

We understand that the low sample size here is concerning. For that reason, we used the Fisher Exact test that compared total phenotype proportions combining the few replicates that could be observed for a single strain. We agree that the data would be better represented as qualitative depictions of phenotype proportions due to the low sample sizes and have removed those statistical comparisons from the figure.

2. I still do not agree that "prokaryotic" is more appropriate than "bacterial." One of the primary criteria used to classify BdpA as a BAR domain protein is based on bioinformatic classification. Are there any similar archaeal proteins? If not, given the number of available archaeal genomes, the eukaryotic/prokaryotic separation seems arbitrary.

See response to essential revision 1 above.

3. My other major concern is whether or not this protein is a BAR-domain protein. That is the main thrust of the paper from the title through much of the text. The effects on membrane stability are interesting but could still be indirect through interactions with the peptidoglycan or LPS sugar components. The homology to eukaryotic BAR domains is borderline. To be clear, this is not a criticism of the authors. They have done a fantastic job given the difficulties of the system. However, I am not convinced that BdpA can be called a BAR domain protein and in several instances (the title, abstract, etc) this claim is centered.

See essential revisions 1 and 2 above.

Reviewer #3:This article identifies in Shewanella oneidensis MR-1 the first prokaryotic BAR domain protein, BdpA, and links its expression to the production of redox-active membrane vesicles and micrometer scale outer membrane extensions (OMEs) in this bacterium. The authors used various complementary approaches to link BdpA expression to the control of OMV size and scaffolding into OMEs. Furthermore, they identified BdpA homologs in other bacteria and heterologously express the protein in BAR-deficient hosts to induce OME production. This research provides much-needed mechanistic understanding of bacterial outer membrane remodeling. I had liked this work very much when I reviewed a previous version of the manuscript. The revisions added clarity to the methods, improved data presentation in the figures, and made for an enjoyable read. I commend the authors for taking the reviewers comments into consideration when revising the manuscript. They did a phenomenal job. I predict this article will establish the intellectual foundation for research in prokaryotic BAR proteins and will be cited extensively.I only have one comment to share in case the authors want to add it to the discussion and/or find it useful for follow-up work. FtsA is related to BdpA and eukaryotic BARs (Figure 5). Given its role in stabilizing the FtsZ ring prior to the assembly of the divisome, FtsA and BdpA could have redundant functions in cell division. This could help explain why the complemented BdpA strain had growth defects (BdpA substitutes for FtsA but may not be able to interact productively with FtsZ or other proteins of the divisome).

See essential revision 2 above.